# Monte Carlo Neural PDE Solver

## Abstract

Training neural PDE solver in an unsupervised manner is essential in scenarios with limited available or high-quality data. However, the performance and efficiency of existing methods are limited by the properties of numerical algorithms integrated during the training stage (like FDM and PSM), which require careful spatiotemporal discretization to obtain reasonable accuracy, especially in cases with high-frequency components and long periods. To overcome these limitations, we propose Monte Carlo Neural PDE Solver (MCNP Solver) for training unsupervised neural solvers via a Monte Carlo view, which regards macroscopic phenomena as ensembles of random particles. MCNP Solver naturally inherits the advantages of the Monte Carlo method (MCM), which is robust against spatial-temporal variations and can tolerate coarse time steps compared to other unsupervised methods. In practice, we develop one-step rollout and Fourier Interpolation techniques that help reduce computational costs or errors arising from time and space, respectively. Furthermore, we design a multi-scale framework to improve performance in long-time simulation tasks. In theory, we characterize the approximation error and robustness of the MCNP Solver on convection-diffusion equations. Numerical experiments on diffusion and Navier-Stokes equations demonstrate significant accuracy improvements compared to other unsupervised baselines in cases with highly variable fields and long-time simulation settings.

## 1   Introduction

Neural PDE solvers, which leverage neural networks as surrogate models to approximate the solutions of PDEs, are emerging as a new paradigm for simulating physical systems with the development of deep learning [33, 31, 23, 12]. Along this direction, several studies have proposed diverse network architectures for neural PDE solvers [30, 33, 5]. These solvers can be trained using supervised [33, 30] or unsupervised approaches [59, 54, 32], employing pre-generated data or PDE information to construct training targets, respectively. The unsupervised training approach is essential for AI-based PDE solvers, particularly in scenarios with limited available or high-quality data. To address this, some studies [54, 32] borrow techniques from classical numerical solvers to construct training targets. For instance, the low-rank decomposition network (LordNet) [54] and physics-informed neural operator (PINO) [32] integrate finite difference method (FDM) and pseudo-spectral methods (PSM) with neural networks during the training stage, respectively. However, FDM and PSM require fine meshes or time steps for stable simulations in general. Therefore, the performance and efficiency of these neural PDE solvers are also limited by the discretization of time and space, particularly when handling highly spatial-temporal variations and simulating physical systems over long periods.

To this end, we propose Monte Carlo Neural PDE Solver (MCNP Solver) for training neural solvers from a Monte Carlo perspective, which regards macroscopic phenomena as ensembles of random movements of microscopic particles [62]. Consequently, for a PDE system with probabilistic representation, MCNP Solver constructs its solutions as training targets via Monte Carlo approximation.

Compared to other unsupervised neural solvers, such as LordNet [54] and PINO [32], MCNP Solver naturally inherits the advantages of MCM. On the one hand, MCNP Solver can tolerate coarse time steps [11, 39], thereby reducing training costs and accumulated errors arising from temporal discretization. On the other hand, it can efficiently handle high-frequency spatial fields due to the derivative-free property of MCM [37, 1]. Moreover, the boundary conditions are automatically encoded into the stochastic process of particles [2, 34], eliminating the need to introduce extra loss terms to satisfy such constraints. In addition to inheriting the benefits of MCM, we also develop one-step rollout and Fourier Interpolation techniques to improve performance and efficiency from the perspective of time and space. Furthermore, we design a multi-scale framework to improve the accuracy and robustness of the MCNP Solver in long-time simulation tasks.

Compared to traditional MCM, MCNP Solver enjoys a significantly faster inference speed once trained. Additionally, traditional MCM requires sampling excess particles to achieve high-precision results, which can lead to severe computational and memory issues. However, thanks to the involvement of neural networks, the MCNP Solver does not necessitate sampling as many particles per epoch during training. According to our experimental observations, the model can converge as expected using gradient descent with only a few particles.

In this paper, we conduct in-depth analyses of the MCNP Solver's performance theoretically and experimentally. In summary, we make the following contributions:

1. We introduce MCNP Solver, a novel Monte Carlo-based unsupervised approach for training neural solvers applicable to PDE systems that allow probabilistic representation. Additionally, we develop several techniques to enhance performance and efficiency, such as Fourier Interpolation, one-step rollout, and multi-scale prediction.

2. Theoretically, we compare the approximation error and robustness of two kinds of neural PDE solvers concerning variations in spatial conditions, temporal discretization steps, and diffusive coefficients. Our theoretical results reveal that MCNP Solver is more robust against the spatial-temporal variants when solving convection-diffusion equations.

3. Our experiments on the diffusion and Navier-Stokes equation (NSE) show significant improvements in accuracy compared to other unsupervised neural solvers for simulating tasks with complex spatial-temporal variants and long-time simulation. Furthermore, the MCNP Solver can obtain comparable or even better results than supervised neural solvers.

## 2 Related Work

**Neural PDE Solver**   Neural PDE solvers have been proposed to learn mappings between functional spaces, such as mapping a PDE's initial condition to its solution [33]. Works like DeepONet [33] and its variants [15, 52, 57, 26] encode the initial conditions and queried locations using branch and trunk networks, respectively. Additionally, Fourier Neural Operator (FNO) [31] and its variants [29, 45, 56] explore learning the operator in Fourier space, an efficient approach for handling different frequency components. Several studies have employed graph neural networks [30, 5] or transformers [6, 28] as the backbone models of neural solvers to adapt to complex geometries. However, these methods require the supervision of ground-truth data generated via accurate numerical solvers, which can be time-consuming in general. To this end, some studies aim to train the neural PDE solvers without the supervision of data [59, 32, 54, 19]. For example, [59] proposed PI-DeepONets, which utilize the PDE residuals to train DeepONets in an unsupervised way. Similarly, [19] proposed Meta-Auto-Decoder, a meta-learning approach to learn families of PDEs in the unsupervised regime. Furthermore, LordNet [54] and PINO [32] borrow techniques from FDM and PSM, and utilize the corresponding residuals as training loss, respectively. Compared to these unsupervised methods, the MCNP Solver incorporates physics information through the Feynman-Kac law, representing a Monte Carlo perspective. This approach allows the solver to efficiently manage diffusion terms, exhibit robustness against spatial-temporal variants, and be suitable for long-time simulations.

**Physics-Informed Neural Networks (PINNs)**   PINNs have been proposed to solve PDE systems by approximating solutions using the PDE residuals, which involve point-to-point mapping between spatial-temporal points and solution values. They are widely employed for solving forward or inverse problems [46, 8, 22, 66]. Recently, PINNs have made significant progress in addressing scientific problems based on PDEs, including NSEs [47, 20, 36], Schrödinger equations [18, 27], Allen Cahn

equations [38, 21], and more. Instead of constructing the loss function directly via the PDE residuals, some works utilize the probabilistic representation to train neural networks [17, 14, 63], which can efficiently handle high-dimensional or fractional PDEs [16, 50, 14, 49, 41]. Furthermore, some studies design loss functions based on other numerical methods, such as the finite volume method [4], finite element method [40, 42], and energy-based method [61]. Notably, the aforementioned PINN methods require retraining neural networks when encountering a PDE with new initial conditions, which can be time-consuming. Moreover, the studies [3, 48] consider PDE families with varying initial conditions while requiring corresponding conditions can be represented by a low-dimensional vector. In this paper, we aim to learn operators between functional spaces that can generalize to different PDE conditions over a distribution. When applying Feynman-Kac laws to this new scenario, we encounter several computational challenges arising from corresponding tasks, such as higher generalization requirements, long-time simulations, and the non-linearity of PDEs. Therefore, we propose Fourier Interpolation, one-step rollout, and multi-scale prediction to overcome these issues. More detailed discussions of these Feynman-Kac-based PINNs can be seen in Appendix D.

# 3 Methodology

## 3.1 Preliminary

In this paper, we consider the general convection-diffusion equation defined as follows:

$$\frac{\partial u}{\partial t} = \boldsymbol{\beta}[u](\boldsymbol{x}, t) \cdot \nabla u + \kappa \Delta u + f(\boldsymbol{x}, t), \quad u(\boldsymbol{x}, 0) = u_0(\boldsymbol{x}), \tag{1}$$

where $\boldsymbol{x} \in \Omega \subset \mathbb{R}^d$ and $t$ denote the $d$-dimensional spatial variable and the time variable, respectively, $\boldsymbol{\beta}[u](\boldsymbol{x}, t) \in \mathbb{R}^d$ is a vector-valued mapping from $u$ to $\mathbb{R}^d$, $\kappa \in \mathbb{R}^+$ is the diffusion parameter, and $f(\boldsymbol{x}, t) \in \mathbb{R}$ denotes the force term. Many well-known PDEs, such as Burgers' equation, NSE, can be viewed as a special form of Eq. 1.

For such PDEs with the form as Eq. 1, the Feynman-Kac formula provides the relationship between the PDEs and corresponding probabilistic representation [43, 44, 16]. In detail, we can use the time inversion (i.e., $\tilde{u}(\boldsymbol{x}, t) = u(\boldsymbol{x}, T - t)$, $\tilde{f}(\boldsymbol{x}, t) = f(\boldsymbol{x}, T - t)$) to the PDE as:

$$\frac{\partial \tilde{u}}{\partial t} = -\boldsymbol{\beta}[\tilde{u}](\boldsymbol{x}, t) \cdot \nabla \tilde{u} - \kappa \Delta \tilde{u} - \tilde{f}(\boldsymbol{x}, t), \quad \tilde{u}(\boldsymbol{x}, T) = u_0(\boldsymbol{x}). \tag{2}$$

Applying the Feynman-Kac formula [35] to the terminal value problem Eq. 2, we have

$$\tilde{u}_0(\boldsymbol{x}) = \mathbb{E}\left[\tilde{u}_T(\tilde{\boldsymbol{\xi}}_T) + \int_0^T \tilde{f}(\tilde{\boldsymbol{\xi}}_s, s)ds\right], \tag{3}$$

where $\tilde{\boldsymbol{\xi}}_s \in \mathbb{R}^d$ is a random process starting at $\boldsymbol{x}$, and moving from 0 to $T$, which satisfies:

$$d\tilde{\boldsymbol{\xi}}_s = \boldsymbol{\beta}[\tilde{u}](\tilde{\boldsymbol{\xi}}_s, s)ds + \sqrt{2\kappa}d\boldsymbol{B}_s, \quad \tilde{\boldsymbol{\xi}}_0 = \boldsymbol{x}, \tag{4}$$

where $\boldsymbol{B}_s$ is the $d$-dimensional standard Brownian motion. Applying time inversion $t \to T - t$ to Eq. 3 and letting $\boldsymbol{\xi}$ be the inversion of $\tilde{\boldsymbol{\xi}}$, we have

$$u_T(\boldsymbol{x}) = \mathbb{E}\left[u_0(\boldsymbol{\xi}_0) + \int_0^T f(\boldsymbol{\xi}_s, s)ds\right]. \tag{5}$$

Furthermore, apart from Eq. 1, some other PDEs can also be handled via the Feynman-Kac formula after certain processing, like wave equations [9] and spatially varying diffusion equations [51].

## 3.2 Monte Carlo Neural PDE Solver

Given a PDE with the form of Eq. 1 and a distribution of the initial conditions $\mathcal{D}_0$, the target of MCNP Solver is to learn a functional mapping $\mathcal{G}_\theta$ with parameter $\theta$ which can simulate the subsequent fields for all initial fields $u_0 \sim \mathcal{D}_0$ at time $t \in [0, T]$. In detail, the inputs and outputs of $\mathcal{G}_\theta$ are given as:

$$\begin{aligned}
\mathcal{G}_\theta : \mathcal{D}_0 \times [0, T] &\to \mathcal{D}_{[0,T]}, \\
(u_0, t) &\mapsto u_t,
\end{aligned} \tag{6}$$

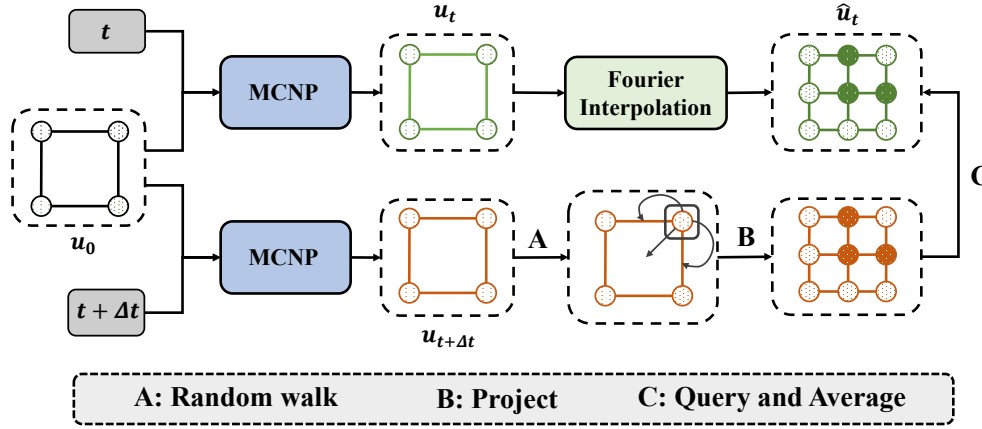

Figure 1: **Illustration of the neural Monte Carlo loss.** We construct the training loss via the relationship between $u_t$ and $u_{t+\Delta t}$ given by the Feynman-Kac law. **A:** random walk according to Eq. 11, and denote the $M$ particles starting at the grid point $\boldsymbol{x}$ as $\{\boldsymbol{\xi}_s^m\}_{m=1}^M$; **B:** when $\boldsymbol{\xi}_s^m$ moving from $t+\Delta t$ to $t$, project each $\boldsymbol{\xi}_t^m$ to the nearest coordinate point $\hat{\boldsymbol{\xi}}_t^m$ in the high resolution coordinate system; **C:** query the value of each $\hat{\boldsymbol{\xi}}_t^m$ via $\hat{u}_t$ and average $\hat{u}_t(\hat{\boldsymbol{\xi}}_t^m)$ as $\sum_{m=1}^M \hat{u}_t(\hat{\boldsymbol{\xi}}_t^m)$. Please note that the high-resolution $\hat{u}_t$ is obtained from $u_t$ via Fourier interpolation. Then, the neural Monte Carlo loss at $\boldsymbol{x}$ is given by: $\|\mathcal{G}_\theta(u_0, t)(\boldsymbol{x}) - \sum_{m=1}^M \hat{u}_t(\hat{\boldsymbol{\xi}}_t^m)\|_2^2$.

where $\mathcal{D}_{[0,T]}$ denotes the joint distribution of the field after $t = 0$. Unlike other supervised operator learning algorithms [27, 33, 5], MCNP Solver aims to learn the operator in an unsupervised way, i.e., only utilize the physics information provided by PDEs. To this end, MCNP Solver considers training the solver via the relationship between $u_t$ and $u_{t+\Delta t}$ (where $0 \le t < t + \Delta t \le T$) derived by the aforementioned probabilistic representation. Considering Eq. 5, an expected neural operator $\mathcal{G}_\theta$ should satisfy the following equation:

$$\mathcal{G}_\theta(u_0, t + \Delta t)(\boldsymbol{x}) = \mathbb{E}_{\boldsymbol{\xi}}\left[\mathcal{G}_\theta(u_0, t)(\boldsymbol{\xi}_t) + \int_t^{t+\Delta t} f(\boldsymbol{\xi}_s, s)ds\right], \tag{7}$$

where $\boldsymbol{\xi}_s(s \in [t, t + \Delta t])$ is the inverse version of stochastic process in Eq. 4 as follows:

$$d\boldsymbol{\xi}_s = -\boldsymbol{\beta}[u](\boldsymbol{\xi}_s, s)ds - \sqrt{2\kappa}d\boldsymbol{B}_s, \quad \boldsymbol{\xi}_{t+\Delta t} = \boldsymbol{x}. \tag{8}$$

Regarding Eq. 7 as the optimization objective, the neural Monte Carlo loss can be written as follows:

$$\mathcal{L}_{\text{MC}}(\mathcal{G}_\theta | u_0, t, \Delta t) = \left\|\mathcal{G}_\theta(u_0, t + \Delta t)(\boldsymbol{x}) - \mathbb{E}_{\boldsymbol{\xi}}\left[\mathcal{G}_\theta(u_0, t)(\boldsymbol{\xi}_t) + \int_t^{t+\Delta t} f(\boldsymbol{\xi}_s, s)ds\right]\right\|_2^2. \tag{9}$$

Equipped with the loss function Eq. 9, we sample the initial states $u_0$ from $\mathcal{D}_0$ and the time $t$ from $[0, T]$ each epoch, and the MCNP loss $\mathcal{L}_{\text{MCNP}}$ is given as follows:

$$\mathcal{L}_{\text{MCNP}} = \mathbb{E}_{u_0 \sim \mathcal{D}_0}[\mathcal{L}_{\text{init}}(\mathcal{G}_\theta | u_0) + \lambda \mathbb{E}_{t \sim [0,T]}[\mathcal{L}_{\text{MC}}(\mathcal{G}_\theta | u_0, t, \Delta t)]], \tag{10}$$

where $\lambda \in \mathbb{R}^+$ is a hyper-parameter, and $\mathcal{L}_{\text{init}}(\mathcal{G}_\theta | u_0) \triangleq \|\mathcal{G}_\theta(u_0, 0) - u_0\|_2^2$ denotes the loss at $t = 0$.

### 3.3 Implementation Details of MCNP Solver

In this section, we introduce some important implementation details for MCNP Solver. We illustrate the framework and training process of MCNP Solver in Fig. 1 and the overall algorithm in Appendix A. We design one-step rollout and Fourier Interpolation trick to reduce the computational cost and error from the perspectives of time and space, respectively. Moreover, we conduct the multi-scale framework to improve the long-time simulation ability of MCNP Solver.

**Temporal Discretization and One-Step Rollout** When simulating the stochastic process in Eq. 8, we utilize the classical Euler–Maruyama method [58] to approximate corresponding SDEs, .i.e,

$$\boldsymbol{\xi}_t = \boldsymbol{\xi}_{t+\Delta t} + \boldsymbol{\beta}[u](\boldsymbol{\xi}_{t+\Delta t}, t + \Delta t)\Delta t + \sqrt{2\kappa}\Delta \boldsymbol{B}_t, \quad \boldsymbol{\xi}_{t+\Delta t} = \boldsymbol{x}. \tag{11}$$

The stochastic integral of the force $f$ in Eq. 7 is approximated via the Euler method, which aligns with [16]. Unlike other Feynman-Kac-based methods [16, 41] conducting random walks in Eq. 8 with multi-steps, we utilize one-step rollout technique to simulate SDEs, i.e., at each $t + \Delta t$, MCNP Solver generates new particles from $\boldsymbol{x}$, and moves them back to $t$ according to Eq. 11. The one-step rollout trick can enforce all $\boldsymbol{\xi}_{t+\Delta t}$ starting at $\boldsymbol{x}$ share the same $\boldsymbol{\beta}[u](\boldsymbol{x}, t + \Delta t)$ during the simulation of SDEs and thus, reduce the computational cost, especially for the scenario when the calculation cost of $\boldsymbol{\beta}$ is expensive. For instance, when the drift $\boldsymbol{\beta}$ term depends on solution $u$, we have to utilize MCNP Solver to calculate $\boldsymbol{\beta}$ accordingly. Moreover, in the NSE conducted in this paper, the mapping $u \to \boldsymbol{\beta}$ represents the transformation from the vorticity field to the velocity field, which involves a numerical integration over an entire domain.

**Random Walks and Boundary Conditions** Eq. 3 and Eq. 4 describe the random walks driven by stochastic processes of corresponding PDEs. For PDEs with periodical boundary conditions, particles should be pulled back according to the periodical law when walking out of the domain $\Omega$. For Dirichlet boundary conditions, the random walk of particles should stop once they reach the boundary. Compared to other unsupervised neural PDE solvers, MCNP Solver encodes the boundary conditions naturally into the random walks of particles and thus does not need additional soft constraints in the loss function. Furthermore, for PDEs with the fractional Laplacian $-(-\Delta)^{\alpha}u$, where $\alpha \in (0, 2)$, we only need to replace the Brownian motion with the $\alpha$-stable Lévy process [24, 65, 64].

**Spatial Discretization and Fourier Interpolation** In this paper, we are interested in the evolution of PDEs at fixed grids $\{\boldsymbol{x}_p\}_{p=1}^{P} \in \Omega$. Consequently, the inputs and outputs of the solver $\mathcal{G}_\theta$ are solution values at $P$ coordinate points. Please note that in Eq .7, the particles $\boldsymbol{\xi}_t$ need to query the value of $\mathcal{G}_\theta(u_0, t)$ when approximating $\mathcal{G}_\theta(u_0, t + \Delta t)$. To efficiently obtain the querying results, we project the locations of particles $\boldsymbol{\xi}_t$ to their nearest neighbor grids in practice. To reduce projection errors, we utilize the Fourier transform to interpolate the fields $u_t = \mathcal{G}_\theta(u_0, t)$ to the high-resolution one $\hat{u}_t$ before the projection. It is worth mentioning that the Fourier Interpolation technique can help the neural solver achieve high-accuracy training signals without the calls of solvers on the high-resolution PDE fields, thereby reducing the training cost.

**Multi-Scale Framework for Long-Time Simulation** When handling tasks with long temporal intervals, we design the following multi-scale framework to make the training process more robust. In detail, we divide the long-time interval $[0, T]$ into $K$ coarse subintervals, i.e., $\{[T_k, T_{k+1}]\}_{k=0}^{K-1}$, with $T_0 = 0$, $T_K = T$ and $T_{k+1} - T_k = \Delta T$. Accordingly, we adopt $K$ neural solvers $\{\mathcal{G}_{\theta_k}\}_{k=0}^{K-1}$ with independent parameter $\theta_k$ to approximate the solution in $[T_k, T_{k+1}]$, respectively. In the training stage, the loss function for long-time simulation is given as follows:

$$\mathcal{L}_{\text{MCNP}}^{\text{Long}} = \mathbb{E}_{u_0 \sim \mathcal{D}_0} \left[ \sum_{k=0}^{K-1} \mathcal{L}_{\text{init}}(\mathcal{G}_{\theta_k} | u_{T_k}) + \lambda \sum_{k=0}^{K-1} \mathbb{E}_{t \sim [T_k, T_{k+1}]} [\mathcal{L}_{\text{MC}}(\mathcal{G}_{\theta_k} | u_{T_k}, t, \Delta t)] \right]. \quad (12)$$

Here, $u_{T_k} = \mathcal{G}_{\theta_{k-1}}(u_{T_{k-1}}, \Delta T)$ can be calculated recursively with $u_{T_0} = u_0$, and $\mathcal{L}_{\text{init}}(\mathcal{G}_{\theta_k} | u_{T_k}) \triangleq \|\mathcal{G}_{\theta_k}(u_{T_k}, 0) - \text{sg}[u_{T_k}]\|_2^2$ denotes the initialization loss for $\mathcal{G}_{\theta_k}$, where $\text{sg}[\cdot]$ denotes the stop-gradient operator. In the inference stage, when predicting the PDE field with the initialization $u_0$ at $t = T_k + \Delta t (0 < \Delta t < \Delta T)$, we first rollout with coarse step $\Delta T$ to obtain $u_{T_k}$, and then adopt finer step to give the prediction of $u_t$ as $\mathcal{G}_{\theta_k}(u_{T_k}, \Delta t)$. Due to the independent parameterization and stop-gradient operator, the proposed multi-scale framework can prevent the prediction at time $t'$ from producing harmful effects on the former time $t < t'$ in the optimization stage. Our experiments reveal that it can improve the performance on long-time simulation tasks where the PDE fields change dramatically over time (e.g., turbulent flow simulation).

# 4 Theoretical Results

In this section, we study the theoretical properties of MCNP Solver when simulating the convection-diffusion equation, and the proof can be seen in Appendix B. In detail, we consider the periodical convection-diffusion equation defined as follows:

$$\frac{\partial u}{\partial t} = \kappa \Delta u + \beta t, \quad x \in [0, 2\pi], \ t \in [0, T], \ \beta \in \mathbb{R}. \quad (13)$$

In the following main theorem, we consider the error of one-step rollout targets provided in PSM and MCM when training neural PDE solvers, respectively.

**Theorem 4.1** *Let $u_t(x)$ be solution of the convection-diffusion equation in the form of Eq. 13, and assume the exact solution at time $t$ can be expressed by the Fourier basis, i.e., $u_t(x) = \sum_{n=1}^{N} a_n \sin(nx)$. Let $\mathcal{G}_\theta$ be the neural PDE solver, and its prediction on $u_t(x)$ can be written as $\mathcal{G}_\theta(u_0, t)(x) = \sum_{n=1}^{N}(a_n + \delta_n)\sin(nx)$, where $\delta_n$ denotes the residual of coefficient on each Fourier basis. Let $H$ and $M$ denote the gird size after Fourier Interpolation and sampling numbers in neural Monte Carlo loss. Let $u_{t+\Delta t}^{\text{PSM}}(x)$ and $u_{t+\Delta t}^{\text{MCM}}(x)$ be the one-step labels starting from $\mathcal{G}_\theta(u_0, t)(x)$, given by PSM and MCM, respectively. Assume $\Delta_t u$ and $u_t(x)$ are Lipschitz functions with respect to $t$ and $x$, respectively, i.e.:*

$$|\Delta_{t_1} u(x) - \Delta_{t_2} u(x)| \le L_{\Delta u}^t |t_1 - t_2|, \quad |u_t(x_1) - u_t(x_2)| \le L_u^x |x_1 - x_2|. \tag{14}$$

*Then, we have*

*1)* $\left| u_{t+\Delta t}^{\text{PSM}}(x) - u_{t+\Delta t}(x) \right| \le \underbrace{\frac{\kappa L_{\Delta u}^t \Delta t^2}{2}}_{\text{E}_1^{\text{PSM}}} + \underbrace{\sum_{n=1}^{N} |\delta_n(\kappa n^2 \Delta t - 1)|}_{\text{E}_2^{\text{PSM}}};$

*2) With probability at least $1 - \frac{(2L_u^x)^2 \kappa \Delta t}{M \epsilon^2}$, we have*

$$\left| u_{t+\Delta t}^{\text{MCM}}(x) - u_{t+\Delta t}(x) \right| \le \underbrace{\frac{1}{2H} \sum_{n=1}^{N} |n a_n|}_{\text{E}_1^{\text{MCM}}} + \underbrace{\sum_{n=1}^{N} |\delta_n|}_{\text{E}_2^{\text{MCM}}} + \underbrace{\epsilon}_{\text{E}_3^{\text{MCM}}} \tag{15}$$

In the PSM, error terms $\text{E}_1^{\text{PSM}}$ and $\text{E}_2^{\text{PSM}}$ arise from the temporal discretization and the perturbation of $\mathcal{G}_\theta(u_0, t)$, respectively. Additionally, the error term $\text{E}_2^{\text{PSM}}$ increases with the rate of $n^2$, where $n^2$ comes from the second order derivative of $\sin(nx)$. To mitigate the error induced by the PSM, one has to decrease $\Delta t$, which inevitably necessitates additional calls to classical or neural solvers. Conversely, for MCM, the error term $\text{E}_1^{\text{MCM}}$ originates from the Fourier Interpolation trick, which can be controlled by increasing the interpolation rate. This operation does not consume much time because it does not require extra solver calls. Moreover, the error caused by the residual $\delta_n$ ($\text{E}_2^{\text{MCM}}$) remains stable as $n$ grows due to the derivative-free property of MCM. It is worth noting that while $\text{E}_3^{\text{MCM}}$ can be controlled by the number of samples $M$, an excessive number of particles is not required in practice. Unlike deterministic biases introduced by other error terms, $\text{E}_3^{\text{MCM}}$ stems from the variance of random processes and can be regarded as a type of stochastic label noise. Some studies [7, 10] have found that such stochastic label noise can aid generalization and even counteract inherent biases. Therefore, we assert that, compared to PSM, the neural Monte Carlo method can tolerate coarser time steps and spatial variations when solving convection-diffusion equations.

# 5 Experiments

In this section, we conduct numerical experiments to evaluate the proposed MCNP Solver on two tasks: 1D diffusion equations and 2D NSEs. Implementation details are introduced in Appendix E. We utilize the FNO [31] as the backbone network, with more detailed discussions in Appendix C. We evaluate the model performance for all tasks via the relative $\ell_2$ error on 200 test PDE samples. We repeat each experiment with three random seeds in $\{0, 1, 2\}$ and report the mean value and variance. All experiments are implemented on an NVIDIA A100 GPU.

## 5.1 1D Diffusion Equation

In this section, we conduct experiments on periodical 1D diffusion equation defined as follows:

$$\frac{\partial u(x,t)}{\partial t} = \kappa \Delta u(x,t), \ x \in [0,1], t \in [0,5]. \tag{16}$$

The initial states $u(x,0)$ are generated from the functional space $\mathcal{F}_N \triangleq \{\sum_{n=1}^{N} a_n \sin(2\pi nx) : a_n \sim \mathbb{U}(0,1)\}$, where $\mathbb{U}(0,1)$ denotes the uniform distribution over $(0,1)$, and $N$ represents the maximum frequency of the functional space.

Table 1: **1D diffusion equation with varying $N$ and $\kappa$.** Relative errors (%) and computational costs for baseline methods and MCNP Solver.

| Model | $\kappa = 0.01$ | | $\kappa = 0.02$ | | Time | | Params |
| :---: | :---: | :---: | :---: | :---: | :---: | :---: | :---: |
| | $N = 6$ | $N = 12$ | $N = 6$ | $N = 12$ | Train (H) | Infer (S) | # (M) |
| PSM | NAN* | NAN | NAN | NAN | – | 0.028 | – |
| PSM+ | 0.000448 | 0.00132 | NAN | NAN | – | 0.554 | – |
| MCM | 5.574± 0.009 | 12.615± 0.056 | 29.991± 0.183 | 83.442± 0.234 | – | 0.034 | – |
| FNO | 1.125± 0.183 | 5.930± 7.468 | 3.662± 0.265 | 23.926± 14.775 | 0.194 | 0.00145 | 0.152 |
| PINO | 1.075± 0.208 | 3.563± 0.684 | 5.275± 2.328 | 26.735± 17.878 | 0.206 | 0.00145 | 0.152 |
| PI-DeepONet | 16.224± 1.165 | 112.630± 18.945 | 113.212± 25.875 | NAN | 2.451 | 0.00126 | 0.153 |
| MCNP | 1.056± 0.194 | 1.511± 0.090 | 3.727± 1.587 | 6.575± 1.948 | 0.116 | 0.00145 | 0.152 |

* Here we unitize NAN to represent the results whose relative error is larger than 200%.

**Experimental Settings** In this setting, $\kappa$ represents the heat transfer rate, with larger $\kappa$ values indicating faster temporal variation rates. $N$ can be regarded as a measure of spatial complexity, where larger values correspond to a higher proportion of high-frequency signals. We select two different $\kappa$ in $\{0.01, 0.02\}$ and $N$ in $\{6, 12\}$, respectively, to evaluate the performance of different methods in handling temporal-spatial variations. We divide the spatial domain $[0, 1]$ into $64$ grid elements for all experiments.

**Baselines** We introduce the baselines conducted on 1D diffusion equations, including: i). **PSM**: A traditional numerical methods. We divide the time interval into 100 uniform lattices and utilize the 2nd Runge-Kutta method for temporal revolution. ii). **PSM+**: PSM with a fine step size. We divide the time interval into 2000 uniform lattices. iii). **MCM**: a traditional numerical method based on the probabilistic representation of PDEs. We set the sampling numbers as $10^5$. iv). **FNO**: Training with 1000 pre-generated data, calculating from the analytic solution of Eq. 16. v). **PINO** [32]: An unsupervised neural operator based on PSM. We divide the time interval into 100 uniform lattices. vi). **PI-DeepONet** [59]: an unsupervised neural operator based on PINN loss and DeepONets. For **MCNP Solver**, we set the sampling numbers and the time step $\Delta t$ as 64 and 0.2, respectively. We interpolate the spatial domain into $1024$ elements in the Fourier Interpolation trick.

**Results** Table 1 presents each method's performance and computational cost on the 1D diffusion equation. Among all unsupervised neural PDE solvers, including PI-DeepONet and PINO, the MCNP Solver performs best on all tasks, particularly for cases with large spatial or temporal variations. Despite PINO obtaining comparable results on the simplest tasks (i.e., $\kappa = 0.01$ and $N = 6$), its error rapidly increases on tasks with $\kappa = 0.02$ or $N = 12$, which is consistent with our theoretical results. The results of PI-DeepONet indicate that the PINN loss cannot efficiently handle high-frequency components, which has also been observed in previous literature [25, 60]. Compared to the supervised method FNO, MCNP Solver obtains comparable results on the tasks when $N = 6$ while significantly outperforming it when $N = 12$, which indicates that more data is required for FNO when handling complex spatial variants. As for classical solvers, PSM fails on all tasks because it requires a fine grid to prevent blowing up, which explains why MCNP Solver can beat PINO. Although PSM+ achieves spectral accuracy on the tasks with $\kappa = 0.01$, it still fails to achieve meaningful results when $\kappa = 0.02$. Moreover, it is more than 380 times slower than other neural solvers due to the refined step size, highlighting one of the main motivations for AI-based PDE studies. MCM's performance is limited by the variance inherent in Monte Carlo simulation, even sampling $10^5$ particles. However, this stochastic label noise arising from the Monte Carlo simulation does not cause apparent harm to the MCNP Solver due to the involvement of neural networks, which is in line with the studies of label noise [7, 10]. In practice, the sampling numbers in MCNP Solver are only set as 64 per epoch, and the neural network can converge as expected with gradient descent during training.

## 5.2 2D Navier-Stokes Equation

In this experiment, we simulate the vorticity field for 2D incompressible flows in a periodic domain $\Omega = [0, 1] \times [0, 1]$, whose vortex equation is given as follows:

$$\frac{\partial \omega}{\partial t} = -(\boldsymbol{u} \cdot \nabla)\omega + \nu \Delta \omega + f(\boldsymbol{x}), \quad \omega = \nabla \times \boldsymbol{u}, \tag{17}$$

Table 2: **2D NSE with varying $\nu$ and $T$.** Relative errors (%) and computational costs for baseline methods and MCNP Solver.

| | Model | Varying $\nu$ | | | Time | | Params |
| --- | --- | --- | --- | --- | --- | --- | --- |
| | | $\nu = 10^{-3}$ | $\nu = 10^{-4}$ | $\nu = 10^{-5}$ | Train (H) | Infer (S) | # (M) |
| $T = 10$ | PSM | 0.309 | NAN | NAN | – | 0.039 | – |
| | PSM+ | 0.103 | 0.136 | 1.521 | – | 0.758 | – |
| | FNO | 1.421$\pm$ 0.068 | 5.155$\pm$ 0.290 | 7.594$\pm$ 0.091 | 0.934 | 0.00255 | 5.319 |
| | PINO | 1.192$\pm$ 0.043 | 5.730$\pm$ 0.046 | 8.952$\pm$ 0.125 | 0.958 | 0.00255 | 5.319 |
| | MCNP | 1.773$\pm$ 0.117 | 4.440$\pm$ 0.157 | 6.539$\pm$ 0.384 | 0.964 | 0.00432 | 4.730 |
| $T = 15$ | PSM | 0.389 | NAN | NAN | – | 0.058 | – |
| | PSM+ | 0.137 | 0.168 | NAN | – | 1.133 | – |
| | FNO | 1.391$\pm$ 0.054 | 5.407$\pm$ 0.103 | 8.429$\pm$ 0.048 | 1.636 | 0.00258 | 7.238 |
| | PINO | 2.161$\pm$ 0.193 | 19.655$\pm$ 5.971 | 24.185$\pm$ 3.947 | 1.703 | 0.00258 | 7.238 |
| | MCNP | 2.195$\pm$ 0.142 | 6.553$\pm$ 0.384 | 8.677$\pm$ 0.350 | 1.458 | 0.00635 | 7.095 |

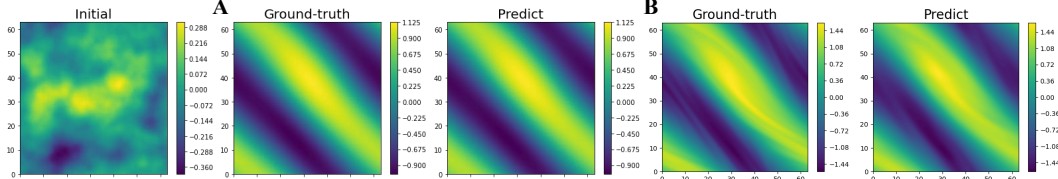

Figure 2: **Simulation of 2D NSE.** The ground-truth solution versus the prediction of a learned MCNP Solver for an example in the test set at $t = 10$, with the viscosity terms $\nu = 10^{-3}$ (**A**) and $\nu = 10^{-5}$ (**B**), respectively.

where $f(\boldsymbol{x}) = 0.1 \sin\left(2\pi\left(\boldsymbol{x}_1 + \boldsymbol{x}_2\right)\right) + 0.1 \cos\left(2\pi\left(\boldsymbol{x}_1 + \boldsymbol{x}_2\right)\right)$ is the forcing function, and $\nu \in \mathbb{R}^+$ represents the viscosity term. The initial vorticity is generated from the Gaussian random field $\mathcal{N}\left(0, 7^{3/2}(-\Delta + 49\boldsymbol{I})^{-2.5}\right)$ with periodic boundaries.

**Experimental Setups**    The viscosity term $\nu$ can be regarded as a measure of the temporal-spatial complexity of NSE. As $\nu$ decreases, the nonlinear term $(\boldsymbol{u} \cdot \nabla)\omega$ gradually governs the motion of fluids, increasing the difficulty of simulation. To evaluate the performance of handling different degrees of turbulence, we conduct the experiments with $\nu$ in $\{10^{-3}, 10^{-4}, 10^{-5}\}$, respectively. We choose two different $T$ in $\{10, 15\}$ to test the long-time simulation ability of each method. We divide the domain $\Omega$ into $64 \times 64$ grid elements.

**Baselines**    We introduce the baselines conducted on 2D NSEs, including:[1] i). **PSM**: We divide the time interval into 100 (150) uniform lattices for $T = 10$ (15) and utilize the Crank–Nicolson scheme for temporal revolution. ii). **PSM+**: We divide the time interval into 2000 (3000) uniform lattices for $T = 10$ (15). iii). **FNO**: Training with 1000 pre-generated data, taking 0.624 hours for data generation. iv). **PINO**: We divide the time interval into 100 and 150 uniform lattices for $T = 10$ and 15, respectively. For **MCNP Solver**, we set the sampling numbers and step size $\Delta t$ to 16 and 0.1, respectively. We interpolate the spatial domain into $256 \times 256$ elements in the Fourier Interpolation trick. The $\Delta T$ in the multi-scale framework is set to 5 for all tasks.

**Results**    Table 2 presents each method's performance and computational cost on the 2D NSEs. As the viscosity term $\nu$ decreases, simulating the flow becomes more challenging for all methods due to increased turbulence, as shown in Fig. 2. Compared to PINO, MCNP Solver achieves comparable results on $\nu = 10^{-3}$ while outperforming it when $\nu = 10^{-4}$ and $10^{-5}$, indicating that MCNP Solver is more accurate on turbulent flow simulation. Furthermore, MCNP Solver has advantages and disadvantages compared to the supervised baseline FNO. On the one hand, MCNP Solver can learn from more training samples due to its data-free regime. On the other hand, the FNO directly uses

---

[1]For PI-DeepONets [59], they only conduct experiments on time-independent PDE in 2D situations in their paper. Furthermore, MCM cannot directly simulate the nonlinear NSE because the unknown velocity $\boldsymbol{u}_{t+\Delta t}$ is required during the simulation of SDE trajectories $\boldsymbol{\xi}_{t+\Delta t} \to \boldsymbol{\xi}_t$.

the ground-truth data as training labels for all $t \in [0, T]$, thus avoiding accumulated errors arising from the calls of the solver during the training stage like other unsupervised methods. As a result, MCNP Solver and FNO achieve better results on most tasks when $T = 10$ and 15, respectively. As for classical solvers, PSM only obtains meaningful results when $\nu = 10^{-3}$, confirming that both PSM and PINO are not robust to coarser time steps. PSM+ achieves the lowest error rate on most tasks but requires almost $180 \sim 300$ times more inference time than other neural solvers.

## 5.3 Ablation Study

We performed several ablation studies of MCNP Solver on NSE ($\nu = 10^{-5}$, $T = 15$) to understand the contribution of each model component. MCNP-OR replaces the one-step rollout technique with two-step when simulating the SDEs. MCNP-FI and MCNP-MS represent the MCNP Solver without the Fourier Interpolation and multi-scale trick, respectively. MCNP-MC replaces the neural Monte Carlo loss with the PSM loss, which aligns with the loss function in PINO. Table 3 reports the results and training costs. MCNP-OR obtains comparable results with MCNP while spending 44% additional training time. Compared to MCNP with MCNP-FI, the Fourier Interpolation trick can significantly improve the accuracy of MCNP while introducing little extra computational cost. The reason is that the rate-determining step in the training stage is the optimization of neural solvers, and the Fourier Interpolation trick does not involve any calls of solvers. Compared to MCNP with MCNP-MS, we can see that the multi-scale framework plays a vital role in improving the long-time simulation ability of MCNP. Additionally, this architecture can reduce the training time because each sub-network is relatively lightweight. Finally, the gap between MCNP and MCNP-MC reveals the advantages of Monte Carlo loss compared to the PSM loss, which is more robust against spatial-temporal variations in turbulence simulation tasks.

Table 3: **Ablation Studies of each model component in MCNP Solver.** Relative error (%) and training time for each method on the NSE tasks with $\nu = 10^{-5}$ and $T = 15$.

|  | MCNP | MCNP-OR | MCNP-FI | MCNP-MS | MCNP-MC |
|---|---|---|---|---|---|
| Error (%) | 8.677± 0.350 | 8.874± 0.150 | 15.561± 0.596 | 24.107± 1.104 | 14.110± 1.789 |
| Time (H) | 1.458 | 2.097 | 1.431 | 2.164 | 1.072 |

## 5.4 Additional Numerical Results

We also conduct experiments to evaluate the MCNP Solver's ability to handle different boundary conditions, fractional Laplacian, and irregular grids, as detailed in Appendix C.

## 6 Conclusion and Discussion

**Conclusion** In this paper, we propose the MCNP Solver, which leverages the Feynman-Kac formula to train neural PDE solvers in an unsupervised manner. Theoretically, we characterize the approximation error and robustness of the MCNP Solver on convection-diffusion equations. Numerical analyses demonstrate the MCNP Solver's ability to adapt to complex spatiotemporal variations and long-time simulations on diffusion equations and NSEs.

**Limitations** This paper has several limitations: (1) The theoretical results are lacking when $\beta$ is not constant, and the gradient flow of the MCNP Solver during the training stage requires further analysis. (2) Some PDEs are not suitable for the Feynman-Kac formula and therefore do not fall within the scope of the MCNP Solver, such as third or higher-order PDEs (involving high-order operators like $u_{xxx}$). (3) The accuracy of the MCNP Solver cannot outperform numerical solvers when disregarding inference time, which is also a major drawback for other existing neural solvers [55, 13]. As discussed in [55], *AI-based methods lack precision compared to classical methods while achieving reasonable accuracy and offering great potential for efficient parameter studies.*

**Future Work** In addition to addressing the limitations, we suggest several directions for future research: (1) Extend the proposed MCNP Solver to broader scenarios, such as high-dimensional PDEs and optimal control problems; (2) Utilize techniques from out-of-distribution generalization [53] to improve the generalization ability of MCNP Solver.

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
