# Appendix: Monte Carlo Neural PDE Solver

## 1 Appendix A: Algorithm Framework

---

**Algorithm 1:** Monte Carlo Neural PDE Solver

---

**Input:** Distribution of initial states $\mathcal{D}_0$, $K$ neural PDE solvers $\{\mathcal{G}_{\theta_k}\}_{k=0}^{K-1}$, time step $\Delta t$ and the coordinates of the fixed grids $\{\boldsymbol{x}_p\}_{p=1}^P$. The time interval $[0, T]$ is divided into $K$ sub-intervals with length $\Delta T$ in the multi-scale framework.

1 **for** *E epochs* **do**
2    - Sample $B$ initial states $\{u_0^b\}_{b=1}^B$ uniformly from $\mathcal{D}_0$;
3    - $\mathcal{L}_{\mathrm{MCNP}} \leftarrow 0$;
4    **for** *k in* $\{0, 1, \cdots, K-1\}$ **do**
5      *% Calculate the initialization loss*;
6      - $\tilde{u}_0^b \leftarrow \mathcal{G}_{\theta_k}(u_0^b, 0)$;
7      - $\mathcal{L}_{\mathrm{MCNP}} \leftarrow \mathcal{L}_{\mathrm{MCNP}} + \sum_{b=1}^B \sum_{p=1}^P \left\| \tilde{u}_0^b(\boldsymbol{x}_p) - u_0^b(\boldsymbol{x}_p) \right\|_2^2$;
8      *% Calculate the MC loss between $t$ and $t + \Delta t$*;
9      - Sample $t$ uniformly in $[0, \Delta T]$ [1];
10      - $\tilde{u}_t^b \leftarrow \mathcal{G}_{\theta_k}(u_0^b, t)$;
11      - Utilize Fourier transform to interpolate the grid of $\tilde{u}_t^b$ to the high resolution one $\hat{u}_t^b$;
12      - Sample $M$ trajectories starting from $\boldsymbol{x}_p$:

$$\boldsymbol{x}_{p,m}^b \leftarrow \boldsymbol{x}_p + \boldsymbol{\beta}[u](\boldsymbol{x}, t + \Delta t)\Delta t + \sqrt{2\kappa}\Delta \boldsymbol{B}_m;$$

13      - Approximate $u_{t+\Delta t}^b$ via the average of $M$ trajectories :

$$u_{t+\Delta t}^b(\boldsymbol{x}_p) \leftarrow \frac{1}{M}\sum_{m=1}^M \hat{u}_t^b(\boldsymbol{x}_{p,m}^b) + f(\boldsymbol{x}_p, t + \Delta t)\Delta t;$$

14      - Calculate the prediction given by $\mathcal{G}_{\theta_k}$: $\tilde{u}_{t+\Delta t}^b = \mathcal{G}_{\theta_k}(u_0^b, t + \Delta t)$;
15      - $\mathcal{L}_{\mathrm{MCNP}} \leftarrow \mathcal{L}_{\mathrm{MCNP}} + \lambda \sum_{b=1}^B \sum_{p=1}^P \left\| \tilde{u}_{t+\Delta t}^b(\boldsymbol{x}_p) - u_{t+\Delta t}^b(\boldsymbol{x}_p) \right\|_2^2$;
16      - Update $u_0^b$: $u_0^b \leftarrow \mathrm{sg}\left[\mathcal{G}_{\theta_k}(u_0^b, \Delta T)\right]$;
17    Update $\mathcal{G}_{\theta_k}$'s parameters: $\theta_k = \mathrm{optim}.\,\mathrm{Adam}(\theta_k, \nabla_{\theta_k}\mathcal{L}_{\mathrm{MCNP}})$ for all $k \in \{0, \cdots K-1\}$;

---

[1]In practice, we sample multiple $t$ in each batch, and the calculation can be conducted simultaneously on GPU.

Submitted to 37th Conference on Neural Information Processing Systems (NeurIPS 2023). Do not distribute.

## Appendix B: Proof of The Main Theorem

In this section, we study the theoretical properties of MCNP Solver when simulating the convection-diffusion equation. In detail, we consider the periodical convection-diffusion equation defined as follows:

$$\frac{\partial u}{\partial t} = \kappa \Delta u + \beta t, \quad x \in [0, 2\pi], \ t \in [0, T], \ \beta \in \mathbb{R}. \tag{1}$$

In the following main theorem, we consider the error of one-step rollout targets provided in PSM and MCM when training neural PDE solvers, respectively.

**Theorem 0.1** *Let $u_t(x)$ be solution of the convection-diffusion equation in the form of Eq. 1, and assume the exact solution at time $t$ can be expressed by the Fourier basis, i.e., $u_t(x) = \sum_{n=1}^{N} a_n \sin(nx)$. Let $\mathcal{G}_\theta$ be the neural PDE solver, and its prediction on $u_t(x)$ can be written as $\mathcal{G}_\theta(u_0, t)(x) = \sum_{n=1}^{N}(a_n + \delta_n)\sin(nx)$, where $\delta_n$ denotes the residual of coefficient on each Fourier basis. Let $H$ and $M$ denote the gird size after Fourier Interpolation and sampling numbers in neural Monte Carlo loss. Let $u_{t+\Delta t}^{\mathrm{PSM}}(x)$ and $u_{t+\Delta t}^{\mathrm{MCM}}(x)$ be the one-step labels starting from $\mathcal{G}_\theta(u_0, t)(x)$, given by PSM and MCM, respectively. Assume $\Delta_t u$ and $u_t(x)$ are Lipschitz functions with respect to $t$ and $x$, respectively, i.e.:*

$$|\Delta_{t_1} u(x) - \Delta_{t_2} u(x)| \le L_{\Delta u}^t |t_1 - t_2|, \quad |u_t(x_1) - u_t(x_2)| \le L_u^x |x_1 - x_2|. \tag{2}$$

*Then, we have*

*1)* $\left| u_{t+\Delta t}^{\mathrm{PSM}}(x) - u_{t+\Delta t}(x) \right| \le \underbrace{\frac{\kappa L_{\Delta u}^t \Delta t^2}{2}}_{\mathrm{E}_1^{\mathrm{PSM}}} + \underbrace{\sum_{n=1}^{N} |\delta_n(\kappa n^2 \Delta t - 1)|}_{\mathrm{E}_2^{\mathrm{PSM}}};$

*2) With probability at least $1 - \frac{(2L_u^x)^2 \kappa \Delta t}{M\epsilon^2}$, we have*

$$\left| u_{t+\Delta t}^{\mathrm{MCM}}(x) - u_{t+\Delta t}(x) \right| \le \underbrace{\frac{1}{2H}\sum_{n=1}^{N} |na_n|}_{\mathrm{E}_1^{\mathrm{MCM}}} + \underbrace{\sum_{n=1}^{N} |\delta_n|}_{\mathrm{E}_2^{\mathrm{MCM}}} + \underbrace{\epsilon}_{\mathrm{E}_3^{\mathrm{MCM}}} \tag{3}$$

**Proof 0.1** *Firstly, we give the upper bound of $\left| u_{t+\Delta t}^{\mathrm{PSM}}(x) - u_{t+\Delta t}(x) \right|$ as follows:*

$$\left| u_{t+\Delta t}^{\mathrm{PSM}}(x) - u_{t+\Delta t}(x) \right|$$
$$= \left| \mathcal{G}_\theta(u_0, t)(x) + \kappa \Delta t \frac{\partial^2 \mathcal{G}_\theta(u_0, t)(x)}{\partial x^2} + b\Delta t - \left[ u_t(x) + \Delta u_t(x)\Delta t + b\Delta t + \kappa \int_t^{t+\Delta t}(\Delta u_s(x) - \Delta u_t(x))\, ds \right] \right|$$
$$= \left| \sum_{n=1}^{N}(a_n + \delta_n)\sin(nx)(1 - \kappa \Delta t n^2) - \sum_{n=1}^{N} a_n \sin(nx)(1 - \kappa \Delta t n^2) - \kappa \int_t^{t+\Delta t}(\Delta u_s(x) - \Delta u_t(x))\, ds \right|$$
$$= \left| \sum_{n=1}^{N} \delta_n \sin(nx)(1 - \kappa \Delta t n^2) - \kappa \int_t^{t+\Delta t}(\Delta u_s(x) - \Delta u_t(x))\, ds \right|$$
$$\le \sum_{n=1}^{N} |\delta_n(\kappa n^2 \Delta t - 1)| + \kappa L_{\Delta u}^t \int_t^{t+\Delta t}(s - t)\, ds$$
$$= \sum_{n=1}^{N} |\delta_n(\kappa n^2 \Delta t - 1)| + \frac{\kappa L_{\Delta u}^t \Delta t^2}{2}. \tag{4}$$

*Please note that PSM estimates the spatial derivative in the Fourier space, and we assume the solution $u$ can be represented by finite basics. Therefore, we ignore the error of the spatial derivative of the PSM in the proof.*

The one-step label constructed in neural Monte Carlo loss can be written as follows:

$$u_{t+\Delta t}^{\mathrm{MCM}}(x) = \frac{1}{M} \sum_{m=1}^{M} \hat{u}_t(x + b\Delta t + \sqrt{2\kappa\Delta t}z_m), \ z_m \sim \mathcal{N}(0,1), \tag{5}$$

where $M$ denotes the number of particles when simulating the stochastic process, $\hat{u}_t$ denotes the solution of $\mathcal{G}_\theta(u_0, t)$ after Fourier Interpolation operation. For any $x \in [0, 2\pi]$, the gap between $\hat{u}_t(x)$ and $u_t(x)$ can be bounded as:

$$
\begin{aligned}
&|u_t(x) - \hat{u}_t(x)| \\
&= \left| \sum_{n=1}^{N} a_n \sin(nx) - \sum_{n=1}^{N} (a_n + \delta_n)\sin(nx') \right| \\
&\leq \sum_{n=1}^{N} |a_n \sin(nx) - \sin(nx')| + \sum_{n=1}^{N} |\delta_n \sin(nx')| \\
&\leq \sum_{n=1}^{N} |na_n||x - x'| + \sum_{n=1}^{N} |\delta_n \sin(nx')| \\
&\leq \frac{1}{2H} \sum_{n=1}^{N} |na_n| + \sum_{n=1}^{N} |\delta_n|,
\end{aligned}
\tag{6}
$$

where $x'$ denotes the nearest grid point to $x$ in the high-resolution coordinate system after Fourier Interpolation operation. Moreover, the variance of $\frac{1}{M}\sum_{m=1}^{M} u_t\left(x + b\Delta t + \sqrt{2\kappa\Delta t}z_m\right)$ can be bounded as follows:

$$
\begin{aligned}
&\mathrm{Var}\left[ \frac{1}{M} \sum_{m=1}^{M} u_t\left(x + b\Delta t + \sqrt{2\kappa\Delta t}z_m\right) \right] \\
&= \frac{1}{M}\mathrm{Var}\left[ u_t\left(x + b\Delta t + \sqrt{2\kappa\Delta t}z\right) \right] \\
&\leq \frac{1}{M} 2(L_u^x)^2 \mathrm{Var}[\sqrt{2\kappa\Delta t}z] \\
&= \frac{(2L_u^x)^2 \kappa \Delta t}{M}.
\end{aligned}
\tag{7}
$$

Thus, according to the Chebyshev's inequality, we have

$$\left| \left[ \sum_{m=1}^{M} \frac{1}{M} u_t(x + b\Delta t + \sqrt{2\kappa\Delta t}z_m) - \mathbb{E}[u_t(x + b\Delta t + \sqrt{2\kappa\Delta t}z)] \right] \right| \leq \epsilon \tag{8}$$

with probability at least $1 - \frac{(2L_u^x)^2 \kappa \Delta t}{M\epsilon^2}$ for any $\epsilon > 0$. Then we can obtain upper bound of $\left| u_{t+\Delta t}^{\mathrm{MCM}}(x) - u_{t+\Delta t}(x) \right|$ with probability at least $1 - \frac{(2L_u^x)^2 \kappa \Delta t}{M\epsilon^2}$ as follows:

$$
\begin{aligned}
&\left| u_{t+\Delta t}^{\mathrm{MCM}}(x) - u_{t+\Delta t}(x) \right| \\
&\leq \left| \frac{1}{M} \sum_{m=1}^{M} \left[ \hat{u}_t(x + b\Delta t + \sqrt{2\kappa\Delta t}z_m) - u_t(x + b\Delta t + \sqrt{2\kappa\Delta t}z_m) \right] \right| \\
&\quad + \left| \left[ \sum_{m=1}^{M} \frac{1}{M} u_t(x + b\Delta t + \sqrt{2\kappa\Delta t}z_m) - \mathbb{E}[u_t(x + b\Delta t + \sqrt{2\kappa\Delta t}z)] \right] \right| \\
&\leq \frac{1}{2H} \sum_{n=1}^{N} |na_n| + \sum_{n=1}^{N} |\delta_n| + \epsilon.
\end{aligned}
\tag{9}
$$

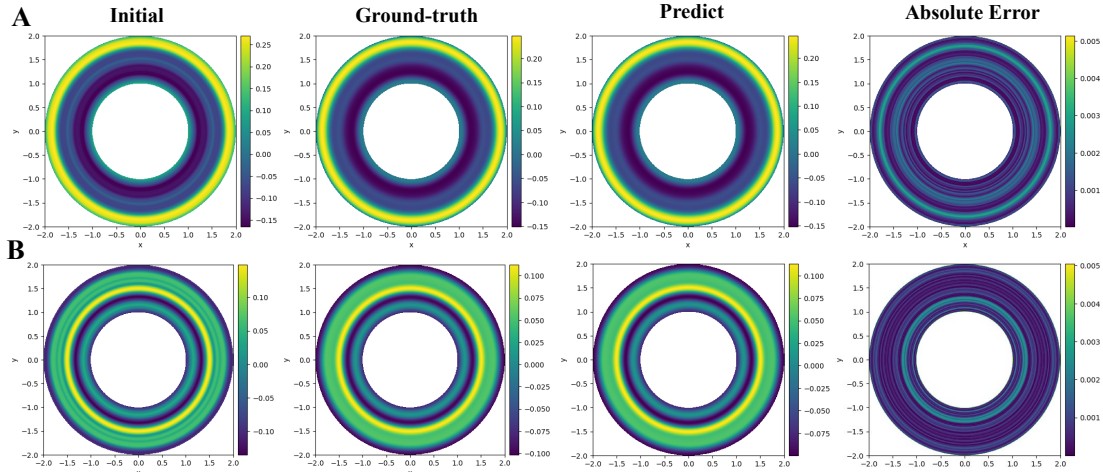

Figure 1: **Simulation of heat diffusion on a circular ring.** The ground-truth solution versus the prediction of a learned NLP Solver for an example in the test set at $t = 1.0s$ with Dirichlet (A) and Neumann (B) boundary conditions, respectively.

Table 1: **Heat diffusion on a circular ring with different boundary conditions.** Relative errors (%) and computational costs for MCM and MCNP Solver.

|  |  | Error (%) | Train Time (H) | Infer Time (S) | # Params (M) |
|---|---|---|---|---|---|
| Dirichlet | MCM | $1.294 \pm 0.0004$ | – | 0.103 | – |
|  | MCNP | $1.222 \pm 0.034$ | 0.235 | 0.00165 | 0.0429 |
| Neumann | MCM | $0.694 \pm 0.0007$ | – | 0.112 | – |
|  | MCNP | $1.211 \pm 0.096$ | 0.241 | 0.00165 | 0.0429 |

## Appendix C: Additional Numerical Results

In this section, we conduct additional experiments to evaluate the MCNP Solver's ability to handle different boundary conditions, fractional Laplacian, and irregular grids. Finally, we study the effects of backbone models.

### C. 1: Heat Diffusion on a Circular Ring

In this experiment, we utilize MCNP Solver to simulate the heat equation on a circular ring, which aims to reveal how MCNP Solver handles different boundary conditions. The center of the ring located at the origin and the radiuses of the two circles are equal to 1 and 2, respectively (Fig. 1). In detail, the PDE is defined as follows:

$$\frac{\partial u(\boldsymbol{x}, t)}{\partial t} = 0.001\Delta u(\boldsymbol{x}, t),$$

$$\text{where } 1 < \|\boldsymbol{x}\|_2^2 < 2, t \in [0, 1]. \tag{10}$$

We consider two different boundary conditions, including Dirichlet and Neumann. For Dirichlet (Neumann) boundary conditions, the random walks of particles need to stop (reflect) when reaching the boundary. As introduced in the main body, the boundary conditions are automatically encoded into the stochastic process of particles [1, 10], eliminating the need to introduce extra loss terms to satisfy such constraints. The initial conditions are set to the spherically symmetric regime; thus, we only need to consider the value of PDEs at $\{(\boldsymbol{x}_1, 0) : \boldsymbol{x}_1 \in [1, 2]\}$. Please note that the random walks of particles are simulated in the 2D space. Fig. 1 shows snapshots of one of the learned heat fields and the corresponding absolute error at $t = 1.0$. Table 1 reveals the performances and computation costs of MCNP Solver and MCM over 200 test instances.

Table 2: **1D fractional diffusion equations with varying** $\alpha$**.** Relative errors (%) and computational costs for MCM and MCNP Solver.

|  |  | Error (%) | Train Time (H) | Infer Time (S) | # Params (M) |
|---|---|---|---|---|---|
| $\alpha = 0.5$ | MCM | $0.540\pm 0.014$ | – | 1.330 | – |
|  | MCNP | $0.410\pm 0.045$ | 0.161 | 0.00157 | 0.152 |
| $\alpha = 1.0$ | MCM | $1.821\pm 0.028$ | – | 0.752 | – |
|  | MCNP | $0.617\pm 0.021$ | 0.145 | 0.00157 | 0.152 |

Table 3: **1D fractional diffusion equations with irregular grids.** Relative errors (%) and computational costs for MCM and MCNP Solver.

|  |  | Error (%) | Train Time (H) | Infer Time (S) | # Params (M) |
|---|---|---|---|---|---|
| $\alpha = 0.5$ | MCM | $0.540\pm 0.014$ | – | 1.545 | – |
|  | MCNP | $0.644\pm 0.013$ | 0.169 | 0.00157 | 0.152 |
| $\alpha = 1.0$ | MCM | $1.942\pm 0.018$ | – | 0.813 | – |
|  | MCNP | $1.095\pm 0.039$ | 0.153 | 0.00157 | 0.152 |

**C.2: 1D Fractional Diffusion Equations**

In this section, we conduct experiments on periodical 1D fractional diffusion equation defined as:

$$\frac{\partial u(x,t)}{\partial t} = -0.01(-\Delta)^{\frac{\alpha}{2}} u(x,t), \ x \in [0,1], t \in [0,5]. \tag{11}$$

Notice that $\alpha = 2$ represents the original Laplacian operator, while $\alpha \in (0,2)$ denotes the fractional operator, which is defined by directional derivatives [12, 9]. We generate the initial states $u(x,0)$ from the functional space $\mathcal{F}_N$ with $N = 12$ in line with Sec. 5.1. We choose two different $\alpha = 0.5$ and 1, respectively. Table 2 reveals the performances and computation costs of MCNP Solver and MCM over 200 test instances. Note that the case $\alpha = 0.5$ takes more inference time for MCM compared with $\alpha = 1.0$ due to the random walk governed by the Lèvy process of $\alpha = 0.5$ needs more computational costs.

**C.3: Irregular Grids: 1D Fractional Diffusion Equations**

MCNP Solver naturally inherits the ability of MCM on handling irregular grids. In this section, we conduct the experiment in Appendix C.2 on irregular grids. We conduct a mapping $f(x) = 1 - \frac{2}{\pi}\arccos(x)$ to transform the uniform grid on $[0,1]$ to the irregular one. Table 3 reveals the performances and computation costs of MCNP Solver and MCM over 200 test instances.

**C.4: The Effects of Backbone Models**

In this section, we discuss the choice of the backbone network of the MCNP Solver. We test three network structures on the 1D diffusion equation ($\kappa = 0.01$ in Sec. 5.1), including FNO [6], Multiwavelet-based Operator (MWT) [4] and UNet [18, 19]. Apart from the above three methods, we also try to utilize the network structure in [2] as a backbone model while failing to obtain meaningful results. The reason might be that the multi-level network structure in [2] is based on MLP, which cannot efficiently handle spatial-temporal variants. To the best of our knowledge, there is no MLP-based model applied in the operator learning tasks. Table 4 reveals the performances and computation costs of each backbone model. According to Table 4, FNO obtains the best performance and efficiency when solving diffusion equations. Therefore, we utilize FNO as the backbone network in this paper. Furthermore, when MCNP Solver uses the FNO as a backbone network, it naturally inherits the corresponding discretization-invariance property [7], i.e., zero-shot super-resolution, as shown in Table 5.

Table 4: **Effects of backbone model.** Relative errors (%) and computational costs for each backbone model.

| | $N = 6$ | $N = 12$ | Train Time (H) | Infer Time (S) | # Params (M) |
|---|---|---|---|---|---|
| MCNP-FNO | $1.056 \pm 0.194$ | $1.511 \pm 0.090$ | 0.116 | 0.00145 | 0.152 |
| MCNP-MWT | $2.103 \pm 0.103$ | $4.810 \pm 0.988$ | 0.492 | 0.0112 | 0.211 |
| MCNP-UNet | $5.148 \pm 1.753$ | $13.248 \pm 4.403$ | 0.813 | 0.00283 | 13.677 |

Table 5: **The discretization-invariance property of MCNP Solver.** Relative error (%) of MCNP Solver trained with grid size 64 via evaluated with $\{64, 128, 256, 512, 1024\}$, respectively.

| size | 64 | 128 | 256 | 512 | 1024 |
|---|---|---|---|---|---|
| $N = 6$ | $1.056 \pm 0.194$ | $1.096 \pm 0.216$ | $1.109 \pm 0.215$ | $1.115 \pm 0.214$ | $1.118 \pm 0.213$ |
| $N = 12$ | $1.511 \pm 0.090$ | $1.543 \pm 0.116$ | $1.559 \pm 0.119$ | $1.567 \pm 0.119$ | $1.571 \pm 0.118$ |

## Appendix D: Other Feynman-Kac (FK)-Based Methods

Some works utilize the probabilistic representation to train neural networks, which mainly focus on the PINN settings with high-dimensional PDEs [5, 17, 16, 11]. The task settings and methodologies of MCNP Solver have remarkable differences from the aforementioned PINN methods, and we list them as follows:

**Generalization requirements** In most FK-based PINN methods, they mainly focus on training a network for one PDE instance and have to retrain the neural network when encountering a PDE with new initial conditions. Moreover, the studies [2, 15] consider PDE families with varying initial conditions while requiring corresponding conditions can be represented by low-dimensional vectors. For MCNP Solver, we aim to learn mappings between functional spaces, and thus the input and output fields are represented by a high-dimensional vector. As a result, FK-based PINN methods mainly utilize MLP-based networks as their backbone model, and we utilize the FNO or other neural operators in the experiments.

**Spatial discretization** When solving the high-dimensional PDEs, the initial fields are usually given by an analytic function. Therefore, the random particles can query the value at any location of $u_0$ without the loss of precision. However, we only can access the value of initial fields at grid points in most settings of low-dimensional PDEs. To reduce the error arising from spatial discretization, we propose a Fourier Interpolation trick to enhance the accuracy of querying.

**Temporal discretization** In other FK-based PINN methods [5, 11], they conduct a multi-step rollout technique when simulating the stochastic process. In MCNP Solver, we utilize the one-step rollout technique to simulate SDEs, i.e., at each $t + \Delta t$, MCNP Solver generates new particles from $x$ and moves them back to $t$. This trick can enforce all $\boldsymbol{\xi}_{t+\Delta t}$ starting at $x$ share the same $\boldsymbol{\beta}[u](x, t + \Delta t)$ during the simulation of SDEs and thus, reduce the computational cost, especially for the scenario when the calculation cost of $\boldsymbol{\beta}$ is expensive (e.g. NSE).

**Long-time simulation** Most FK-based methods are interested in the tasks with short-time simulations [5, 17, 16, 11]. The final time $T$ in their experiments is less than 1 in general. However, in low-dimensional tasks, it is important to simulate the fluids or heat flows for a long-time in realistic scenarios. As the results of ablation studies shown in Sec. 5.3, plain network structures can lead to unstable simulation for long-time tasks. It is worth mentioning that some studies [5] also divide the time interval $[0, T]$ into several sub-intervals $[t.t + \Delta t]$ with small $\Delta t$, where $\Delta t$ is the step size of MCM when simulating the corresponding SDEs. Then, they utilize neural networks with different parameters to solve the PDE in each $[t, t + \Delta t]$. However, when transferring this technique directly to the long-time simulation can arise severe computational and memory issues. In this work, we utilize the multi-scale framework, which divides the time interval $[0, T]$ into $K$ coarse time interval, whose length $\Delta T$ is much longer than the $\Delta t$. We construct the initialization loss and the neural Monte Carlo loss on the coarse and fine intervals, respectively. According to our numerical results, the multi-scale framework can enhance the robustness and efficiency of the MCNP Solver.

## Appendix E: Implementation Details

### E.1: Baselines

In this paper, we adopt Pytorch [13] to implement MCNP Solver, FNO, and PINO, and JAX [3] for PI-DeepONet, respectively. Here, we introduce two different unsupervised methods as follows.

**PI-DeepONet [21]**  PI-DeepONet utilized the PDE residuals to train DeepONets in an unsupervised way. The loss function in PI-DeepONet can be formulated as follows:

$$\mathcal{L}_{\text{PI-DeepONet}} = \mathcal{L}_{\text{operator}} + \lambda \mathcal{L}_{\text{physics}},$$
$$\text{where} \quad \mathcal{L}_{\text{operator}} = \text{MSE}[\mathcal{G}_\theta(u_0^b, t = 0)(\boldsymbol{x}_p) - \mathcal{G}(u_0^b, t = 0)(\boldsymbol{x}_p)], \tag{12}$$
$$\mathcal{L}_{\text{physics}} = \text{MSE}[\mathcal{R}(\mathcal{G}_\theta(u_0^b, t)(\boldsymbol{x}_p), \boldsymbol{x}_p, t)],$$

where MSE represents the mean square error, $\mathcal{G}_\theta$ represents a neural operator, $\mathcal{G}$ and $\mathcal{R}$ denote the ground-truth and the residual of the PDE operator, respectively. As shown in Eq. 12, $\mathcal{L}_{\text{operator}}$ and $\mathcal{L}_{\text{physics}}$ enforce $\mathcal{G}_\theta$ to satisfy the initial conditions (or boundary conditions) and the PDE constraints, respectively. Like PINNs [14], the PDE residuals in Eq. 12 are calculated via the auto-differentiation.

**PINO [8]**  PINO utilized the PSM to construct the loss function between $\mathcal{G}_\theta(u_t^b)$ and $\mathcal{G}_\theta(u_{t+\Delta t}^b)$, and PINO utilized the FNO [6] as the backbone network. The loss function in PINO can be formulated as follows:

$$\mathcal{L}_{\text{PINO}} = \mathcal{L}_{\text{operator}} + \lambda \mathcal{L}_{\text{physics}},$$
$$\text{where} \quad \mathcal{L}_{\text{operator}} = \text{MSE}[\mathcal{G}_\theta(u_0^b, t = 0)(\boldsymbol{x}_p) - \mathcal{G}(u_0^b, t = 0)(\boldsymbol{x}_p)],$$
$$\mathcal{L}_{\text{physics}} = \sum_{t=0}^{T-\Delta t} \text{MSE}[\mathcal{G}_\theta(u_0^b, t + \Delta t)(\boldsymbol{x}_p) - \mathcal{G}_\theta(u_0^b, t)(\boldsymbol{x}_p) - \mathcal{P}(\mathcal{G}_\theta, \boldsymbol{x}_p, t)], \tag{13}$$

where $\mathcal{P}$ denotes the update regime of PSM.

### E.2: 1D Diffusion Equation

**Data**  We conduct experiments on periodical 1D diffusion equation defined as follows:

$$\frac{\partial u(x,t)}{\partial t} = \kappa \Delta u(x,t), \; x \in [0,1], t \in [0,5]. \tag{14}$$

The initial states $u(x,0)$ are generated from the functional space $\mathcal{F}_N \triangleq \{\sum_{n=1}^{N} a_n \sin(2\pi nx) : a_n \sim \mathbb{U}(0,1)\}$, where $\mathbb{U}(0,1)$ denotes the uniform distribution over $(0,1)$, and $N$ represents the maximum frequency of the functional space. The data is generated via the following exact solution of Eq. 14:

$$u(x,t) = \sum_{n=1}^{N} a_n \sin(2\pi nx) e^{-\kappa(2\pi n)^2 t}. \tag{15}$$

We generate 1000 training data with seed 1, and 200 test data with seed 0.

**Hyperparameters**  We first conduct experiments on the supervised tasks to search for the best network structure of 1D FNO. We fix the number of layers as 4 and choose the best $width$ in $\{10, 20, 30\}$ and $mode$ in $\{12, 16, 20, 24\}$ for FNO, respectively. As a result, the 4-layer 1D FNO with $width = 30, mode = 20$ obtains the best performance, and we utilize it as a backbone model in all FNO-based experiments. For FNO, we utilize Adam to optimize the neural network for 2000 epochs with the initial learning rate of 0.02 and decay the learning rate by a factor of 0.5 every 500 epochs. The batch size is fixed as 200. The learning rate is chosen from the set $\{0.02, 0.01, 0.005\}$. For PINO, we utilize Adam to optimize the neural network for 10000 epochs with an initial learning rate of 0.01 and decay the learning rate by a factor of 0.5 every 500 epochs. The batch size and $\lambda$ are fixed as 200 and 0.01. The learning rate and $\lambda$ are chosen from the set $\{0.02, 0.01, 0.005\}$ and $\{0.1, 0.05, 0.01\}$. For MCNP Solver, we utilize Adam to optimize the neural network for 10000 epochs with the initial learning rate of 0.01 and decay the learning rate by a factor of 0.5 every 500

148 epochs. The batch size and $\lambda$ are fixed as 200 and 0.1. The learning rate and $\lambda$ are chosen from the set
149 $\{0.02, 0.01, 0.005\}$ and $\{0.1, 0.05, 0.01\}$. For PI-DeepONet, we choose the network structure in line
150 with the 1D case in [21], and extend the training iterations to 200000 to make sure the convergence
151 of the model. Moreover, we search the $\lambda$ in $\{0.001, 0.01, 0.1, 0.2, 0.5, 1\}$ and fix it as 0.2.

## E.3: 2D Navier-Stokes Equation

153 **Data**   We utilize the PSM to generate the ground truth test data with the time-step of $10^{-4}$ for the
154 Crank–Nicolson scheme. Furthermore, all PDE instances are generated on the grid $256 \times 256$, then
155 downsampled to $64 \times 64$, which is in line with the setting in [6]. We generate 1000 training data with
156 seed 1, and 200 test data with seed 0.

157 **Hyperparameters**   We first conduct experiments on the supervised tasks to search for the best
158 network structure of 2D FNO. We fix the number of layers as 4 and choose the best $mode$ in
159 $\{12, 16, 20, 24\}$ for FNO. As a result, the 4-layer 2D FNO with $mode = 16$ obtains the best
160 performance. We set $width$ as 36 and 42 for the tasks with $T = 10$ and 15, respectively. And
161 the $width$ for MCNP Solver is fixed as 24. Due to the multi-scale framework in MCNP Solver,
162 all methods have comparable model sizes. For FNO, we find that a cosine annealing schedule can
163 obtain the best result when training with the supervised regime. Therefore, we utilize Adam to
164 optimize the neural network for 200 epochs with the initial learning rate of 0.01, and decay with
165 cosine annealing schedule ($T_{\max} = 20$). The batch size is fixed as 20. The learning rate is chosen
166 from the set $\{0.02, 0.01, 0.005\}$. For PINO, we utilize Adam to optimize the neural network for
167 10000 epochs with the initial learning rate of 0.005 and decay the learning rate by a factor of 0.5 every
168 2000 epochs. The batch size and $\lambda$ are fixed as 16 and 0.1. The learning rate and $\lambda$ are chosen from
169 the set $\{0.02, 0.01, 0.005\}$ and $\{0.1, 0.05, 0.01\}$. For MCNP Solver, we utilize Adam to optimize
170 the neural network for 10000 epochs with the initial learning rate of 0.01 and decay the learning rate
171 by a factor of 0.5 every 500 epochs. The batch size and $\lambda$ are fixed as 200 and 0.05. The learning rate
172 and $\lambda$ are chosen from the set $\{0.02, 0.01, 0.005\}$ and $\{0.1, 0.05, 0.01\}$.

## E.4: Heat Diffusion on a Circular Ring

174 **Data**   The ground-truth data is generated via the Python package 'py-pde' [22], and the step size is
175 fixed as $10^{-4}$. The initial heat distribution is generated from $u_0 \sim \mathcal{N}\left(0, 3^{3/2}(-\Delta + 9I)^{-1}\right)$, and
176 the width of the ring is divided into 256 lattices.

177 **Hyperparameters**   In this experiment, we utilize the 4-layer 1D FNO as the backbone network,
178 with $width = 20, mode = 12$ and GeLU activation. We utilize Adam to optimize the neural operator
179 for 10000 epochs with an initial learning rate of 0.01 and decay the learning rate by a factor of 0.5
180 every 500 epochs. For each epoch, we sample 200 initial conditions from $\mathcal{D}_0$ and 16 particles to
181 simulate the random processes. We set the time step $\Delta t$ and $\lambda$ as 0.05 and 0.1. For MCM, we set the
182 time step $\Delta t$ and the sampling numbers as 0.05 and $10^4$.

## E.5: 1D Fractional Diffusion Equations

184 **Data**   The data is generated via the following exact solution of Eq. 11:

$$u(x, t) = \sum_{n=1}^{N} a_n \sin(2\pi n x) e^{-\kappa(2\pi n)^\alpha t}. \tag{16}$$

185 The spatial field is divided into 128 lattices. We generate 1000 training data with seed 1, and 200 test
186 data with seed 0.

187 **Hyperparameters**   In this experiment, we utilize the 4-layer 1D FNO as the backbone network,
188 with $width = 30, mode = 20$ and GeLU activation. We utilize Adam to optimize the neural operator
189 for 10000 epochs with an initial learning rate of 0.01 and decay the learning rate by a factor of
190 0.5 every 500 epochs. For each epoch, we sample 200 initial conditions from $\mathcal{D}_0$ and 64 particles
191 to simulate the random processes. We set the time step $\Delta t$ and $\lambda$ as 0.2 and 0.01. For MCM, we
192 set the time step $\Delta t$ and the sampling numbers as 0.05 and $10^4$. Furthermore, we need to mention
193 that there is no GPU package for the Lévy sampling as far as we know. Thus, we utilize the code
194 $\mathrm{scipy.stats.levy\_stable}$ [20] to generate the corresponding random processes.