# OpenReview forum: "Monte Carlo Neural PDE Solver"
_NeurIPS.cc/2023/Conference — Submitted to NeurIPS 2023_

### Official Review · Reviewer_MGTF · 2023-07-01

**Soundness:** 1 poor
**Presentation:** 1 poor
**Contribution:** 2 fair
**Rating:** 3
**Confidence:** 4

**Summary:**

The authors propose an unsupervised neural technique for solving PDEs based on the classical correspondence between (parabolic) partial differential equations (PDE) and stochastic differential equations (SDE) as given by the Feynman-Kac formula. Specifically, they propose minimizing the error between the neural approximation's deterministic prediction at timestep t+1 and the expected prediction of the neural approximation over particles that have evolved up to time t stochastically according to the Feynman-Kac SDE representation of the given PDE.

**Strengths:**

Incorporating Feynman-Kac into the PINN framework is an interesting idea.

**Weaknesses:**

# Poor numerics
The experiments do not justify the claims, e.g. that long rollouts are more stable using this method. To prove this, at the very least the authors need to present results where the trajectories actually exhibit turbulence. Then, an ablation with the multi-scale framework is required.

# Poor presentation
The authors cannot expect the reader to be familiar with Feynman-Kac and need to give explanations in plain English of the significance of this result. Then, the equations in the main paper should aim to clarify this further, not give a comprehensive mathematical presentation. For example, the inclusion of the forcing function  distracts from the main result which is using time reversal to obtain an SDE that moves in the right time direction for equations that are most often solved using neural networks, e.g. ones with an initial condition, not a final condition.

Furthermore, only the most experienced reads will walk away from this paper with a clear idea of how to implement the proposed algorithm. The emphasis in the paper should be to give the reader something to implement and try, not a theoretical proof.

Finally, there are a number of tricks that are not included in the initial idea and are not sufficiently explained. For example, the Fourier interpolation. The multi-scale framework makes the model non-parametric which is a significant departure from previous work and from the presentation of this paper as a Monte-Carlo approximation to PDEs.

**Questions:**

Please explain the Fourier interpolation, that would be very helpful.

---

> ### Author Rebuttal · Authors · 2023-08-09
>
> Thank you for your time and feedback! Based on your comments, we provide some replies to address your concerns as follows:
>
> **The trajectories of NSE.**
>
> Thanks for your suggestion! We add figures in the PDF of the **Global Response B** part to show the variation of trajectories and errors for each neural solver over time.
>
> **An ablation with the multi-scale framework.**
>
> Thanks for your comments! The ablation studies can be seen in Sec. 5.3, where the effects of all tricks are evaluated.
>
> **The equations should aim to clarify this further, not give a comprehensive mathematical presentation.**
>
> Thank you for your feedback! We follow the common description of Feymann-Kac law like most of the related literatures do [1, 2, 3]. As PDE belongs to mathematics, it is unavoidable to leverage advanced mathematical tools, and we believe that the NeurIPS readers in this research field could follow it if they are interested in neural PDE solvers or neural operators. Compared to other Feymann-Kac-based papers [1, 2, 3], we have tried to simplify the mathematical formulas in this paper, and we give an intuitive explanation in Lines 36-37, _the probabilistic expression of the PDE regards macroscopic phenomena as ensembles of random movements of microscopic particles._ Furthermore, considering that most PDEs are given by forward form, we introduce how to use time reversal to transform the initial value problem into the final value problem, rather than directly giving the final value form [1, 2, 3]. For readers who are not familiar with the Feynman-Kac formula, they may need to acknowledge the equivalence between the PDE (Eq. 3) and the corresponding SDE (Eq. 4) by default, and then the subsequent reading may not be affected too much.
>
> We will seriously consider your suggestions and try to make corresponding modifications in the final version. Any further detailed comments on improving the presentation will be welcomed and helpful.
>
>
> **Authors should give the reader something to implement and try, not theoretical proof.**
>
> Thank you for your suggestions! We show the overall algorithm framework in Appendix A, and upload the code (please refer to the **Global Response C** part). We believe the readers can understand the implementation detail of the MCNP according to these materials. Furthermore, we hope the theoretical results can help the readers understand the advantages of MCNP when handling PDEs with large spatiotemporal variants. However, due to the page limit, we cannot show both the theoretical results and the algorithm framework in the main body. If you think replacing the theorem with the algorithm framework would be better, we will do so in the final version.
>
> **The Fourier interpolation trick.**
>
> Thanks for your question! We utilize Fourier transform to map the $N$ PDE fields with low resolution (like $N \times 64$, where $N$ is the batch size) to the frequency domain, and use the inverse Fourier transform to remap it to the high-resolution space (like $N \times 256$). The Fourier interpolation trick can be conducted in one line with the help of PyTorch as follows:
>
> ```python
> # u.shape = (N, 64)
> u_super = 4 * torch.fft.irfft(torch.fft.rfft(u), n=256) # u_super.shape = (N, 256)
> ```
>
> **The multi-scale framework.**
>
> In this paper, we aim to train neural PDE solvers in an unsupervised manner, especially in the scenario with high-frequency components. In such a scenario, stable and long-term simulating PDE fields would be challenging due to the absence of supervised data and the large deformation of PDEs. Therefore, we propose a multi-scale framework to enhance the robustness of MCNP, which is indeed less relevant to Feymann-Kac law, but plays an important role in the tasks targeted in this paper (Please refer to Sec. 5.3 for ablation studies).
>
> We hope our rebuttal can address your concerns. Also, we would like to know whether there are any other questions, and we are happy to answer and discuss them. If the major concerns have been addressed, could you please kindly raise the rating?
>
> [1] Han J, Jentzen A, E W. Solving high-dimensional partial differential equations using deep learning. Proceedings of the National Academy of Sciences, 2018.
>
> [2] Richter L, Berner J. Robust SDE-based variational formulations for solving linear PDEs via deep learning. ICML, 2022.
>
> [3] Berner J, Dablander M, Grohs P. Numerically solving parametric families of high-dimensional Kolmogorov partial differential equations via deep learning. NeurIPS, 2020.

---

### Official Review · Reviewer_urJf · 2023-07-04

**Soundness:** 2 fair
**Presentation:** 4 excellent
**Contribution:** 4 excellent
**Rating:** 6
**Confidence:** 5

**Summary:**

The authors present MCNP, a new unsupervised training loss for surrogate simulation networks. This loss is based on the link between stochastic processes and PDEs, sampling one-step Brownian motion to estimate the PDE solution. The learned network takes an initial state and the target simulation time to compute the state at that time in one pass. For longer periods, multiple NNs are trained, one for each sub-interval.

**Strengths:**

The paper is generally well-written and relatively easy to understand. It includes a good overview over related work.

The paper includes both theoretical and numerical results. The method is derived using the Feynman-Kac formula and the authors show how the errors of PSM and MCM scale when given an incorrect input state, such as predicted by a neural network.

A total of five numerical experiments are performed, covering a large range of simulation configurations. The paper contains an ablation study, giving some insight into the impacts of various parts of the MCNP method.

**Weaknesses:**

While the experiments are varied in the tested configurations, all but on experiment consider simple diffusion equations, some of which can be solved analytically.

The paper does not show any simulation trajectories and gives no insight into how the various tested methods behave in their experiments. Instead, only the final losses are reported. This makes it hard to determine the cause of improvement from the numerical results. I strongly recommend documenting your observations in the appendix.

The Navier-Stokes experiment seems to be mostly forcing-driven with all initial states ending in a similar configuration. A different forcing, such as Kolmogorov flow, would result in a much wider range of trajectories.

The paper does not give details as to how the numerical simulation (PSM) of the Navier-Stokes experiment was performed. Please describe the simulator in more detail if you implemented it yourself.

The source code is not part of the submission and the authors have not declared their intent to make it public. I strongly recommend doing so, especially if the employed CFD solver was implemented from scratch.

Minor:

* Eq. 3: Please indicate the x-dependence of xi
* Fig 1 caption: The formulas are missing a factor of 1/M
* L184: You mention that cutting the gradients prevents numerical instabilities but ignore the positive effects that gradient backpropagation can have.
* You refer to Eq.13 as a convection-diffusion equation but it does not contain a convection term.
* L199: The notation Δ ₜ u is confusing since Δ is already in use.
* L216: The argument that label noise helps MCNP perform coarser time steps seems unfounded to me. This requires an explanation.
* L237: By lattices, you probably mean frames or time steps?
* L268 and Eq. 17: You can simplify the forcing to a single sine term.
* Figure 2 is never referenced.
* Figure 2 shows the vorticity, correct? Please specify in the caption.

**Questions:**

You state that the random walk of particles should stop when they hit a Dirichlet boundary. Is this still true when performing multiple steps for the time integration of the stochastic process? I think this would over-value the boundary condition, as particles are denied the chance to re-enter the domain.

Using the Feynman-Kac formula, you could also derive a formulation where the stochastic process is simulated forward in time instead of backward. Have you tried this?

In your periodic diffusion experiments, PSM should be equal to a purely spectral method, right?

Why did you choose a vorticity formulation for the Navier-Stokes experiment instead of a velocity formulation?

How did you choose the temporal discretization (like 100 for PSM, 2000 for PSM+) for the tested methods?

The Fourier trick seems to emulate interpolation on the grid. Have you tried linear interpolation or Fourier-upsampling + linear interpolation?

You say that you run some baselines with PyTorch and others with Jax. Is JIT compilation enabled on all examples? How big do you think the performance difference would be when switching all to one library?

It looks like the PSM methods fail due to the time increment being too large. Have you looked at the CFL numbers? Would this problem be resolved by dynamic time steps?

**Limitations:**

The limitations are adequately discussed.

---

> ### Author Rebuttal · Authors · 2023-08-09
>
> Thanks for your valuable feedback! According to your constructive comments, we make some replies as follows:
>
> **W1 & W3. The experiments.**
>
> To address your concerns, we added an experiment to simulate Kolmogorov flow (see **Global Response. B**). Furthermore, we chose PDEs with analytical solutions to avoid the bias of numerical solvers. For example, PINO is based on PSM, and it would be unfair to MCNP if the ground truth is generated via PSM. Using PDE with analytical solutions is also a common approach to evaluate the performance in computational mathematics [1, 2].
>
> **W2. Simulation trajectories.**
>
> Thanks for your suggestions! We add figures in the PDF of **Global Response** to show the variation of trajectories and errors over time.
>
> **W4, Q3, Q5 & Q8. Implementation of PSM.**
>
> We use the code in the original neural PDE paper [3], whose implementation is a standard benchmark in neural PDE papers. For diffusion equation, the PSM is equal to the purely spectral method. For temporal discretization, we set PSM to align with the time step of PINO and hope to reveal why PINO fails. For PSM+, we refine its time step to make it succeed on most problems. We acknowledge that the PSM can outperform all ML-based methods if we further refine its grid, and some advanced tricks could enhance the performance, like dynamic time steps. In this paper, we only consider the standard implementation in most operator learning papers [3, 4]. And even compared with the PSM conducted on the coarse grid, neural solvers still enjoy a 20 times speed up.
>
> **W5. Source code.**
>
> We have uploaded the code (please refer to **Global Response. C**).
>
> **Minor Problems**
>
> Please note that we have updated the theorem parts in the **Global Response. A**. The lattices mean the time steps in this paper. Fig. 2 represents the vorticity.
>
> For label noise, let's consider learning the network with the noisy label $y=y_g+e_d+e_r$, where $y_g$ is the ground truth, $e_d$ is the deterministic bias, and $e_r$ is random noise, which could change per epoch during training. Some papers discovered that $e_{r}$ could help the training, even counteracts $e_d$ [5]. We regard the deterministic error and the random error brought by MCM as $e_d$ and $e_r$, respectively. However, we acknowledge that rigorously proving still needs further analysis, and the main challenge is the nonconvex property of neural networks. To demonstrate this empirically, we add an experiment to reveal the effects of M on MCNP (settings align with Sec. 5.1, $\kappa$=0.02).
>
> |M|32|64|128|256|
> |-|-|-|-|-|
> |Error, N=6|3.467$\pm$ 0.470|3.727$\pm$ 1.587|3.543$\pm$ 1.633|3.648$\pm$ 1.222|
> |Error, N=12|6.322$\pm$ 0.991|6.575$\pm$ 1.948|6.564$\pm$ 1.902|8.731$\pm$ 2.738|
> We can see that:
> - The results of MCNP are relatively robust to M.
> - M is not necessarily better when larger, which indicates that controlling the noise within a certain level can help generalization.
>
> Thanks for your feedback! We will clarify them in the final version.
>
> **Q1. Dirichlet boundary.**
>
> We stop the random walk when the particle hits the boundary, which is consistent with the mathematical principle of the Feymann-Kac theorem. A more detailed mathematical explanation can be found in lecture note [6] (Theorem 4.2.1)
>
> **Q2. Simulation of SDE.**
>
> In theory, the SDE is given by a **backward** form in Feymann-Kac law [6] (Theorem 4.1.2). In practice, we try to simulate the SDE with a forward formula but fail to obtain meaningful results.
>
> **Q4. Vorticity formulation.**
>
> We choose this formulation to align with the settings in other neural PDE papers [3, 4].
>
> **Q6. Fourier Interpolation.**
>
> Before writing this paper, we tried other interpolation tricks, including linear and bilinear interpolation. However, Fourier interpolation obtained the best performance. One potential reason is that Fourier transform is more compatible with the characteristics of PDE.
>
> **Q7. PyTorch v.s. Jax.**
>
> In this paper, we only use Jax to conduct the experiments of PI-DeepONet due to the absence of the PyTorch code. In the code of PI-DeepONet, JIT compilation is involved, while other PyTorch methods are not. According to some literature, the main difference between deep learning APIs is speed and memory [7]. As reported in [8], _the JAX implementation is about 2.5-3.4x faster than PyTorch! However, with larger models, larger batch sizes, or smaller GPUs, the speed-up is expected to become considerably smaller._ In this paper, we use PyTorch as the main API because it is the most popular API in operator learning papers, and we hope to align the experimental settings with other papers.
>
> We hope our rebuttal can address your concerns. Also, we would like to know whether there are any other questions, and we are happy to answer and discuss them. If the major concerns have been addressed, could you please kindly raise the rating?
>
> [1] Labovsky A E. Approximate deconvolution with correction: A member of a new class of models for high Reynolds number flows. SIAM Journal on Numerical Analysis, 2020.
>
> [2] Li B, Zhang J, et al. Stability and error analysis for a second-order fast approximation of the one-dimensional Schrodinger equation under absorbing boundary conditions. SIAM Journal on Scientific Computing, 2018.
>
> [3] Li Z, Kovachki N, et al. Fourier neural operator for parametric partial differential equations. ICLR, 2020.
>
> [4] Wu T, Maruyama T, et al. Learning to accelerate partial differential equations via latent global evolution. NeurIPS, 2022.
>
> [5] Chen P, Chen G, et al. Noise against noise: stochastic label noise helps combat inherent label noise. ICLR, 2020.
>
> [6] Nolen J. Partial differential equations and diffusion processes. Lecture Notes, Stanford University, 2009.
>
> [7] Paszke A, Gross S, et al. Pytorch: An imperative style, high-performance deep learning library. NeurIPS, 2019.
>
> [8] Tutorial 5 (JAX): Inception, ResNet and DenseNet.

---

> > ### Comment · Reviewer_urJf · 2023-08-12
> > **Some further questions**
> >
> > Thank you for answering my questions and providing additional experiments.
> >
> > The inclusion of Kolmogorov flow and example trajectories strengthens the paper. I hope you can also provide trajectories for the other experiments in the appendix and plot more than just one example per experiment.
> >
> > Your discussion still does not explain in what way your method leads to more stable inferred trajectories. I realize that a detailed analysis of the advantages and disadvantages is not something you can do in a week but still I’d appreciate any insight you can give into why and how your method performs better than each of the baselines. What kind of mistakes do the different methods tend to make?
> >
> > In Fig. 2 of the attached PDF page, what is going on with FNO in the case $\nu=10^{-3}$? Also, the error metric does not seem to match Table 2 from the main paper. E.g. for $\nu=10^{-4}$ and $T=15$, Table 2 claims a relative error of 6.553% for MCNP while the diagram shows about 14%.
> >
> > I will update my rating after discussing with the other reviewers and the AC.
> >
> > *Side note:* As a reviewer, I assign my ratings as objectively as I can. Of course, I will take the rebuttal into account. However, I do not appreciate being asked by the authors to raise my rating. I’d hate for OpenReview to turn into a platform where authors must beg for their scores to be raised.

---

> > > ### Author Response · Authors · 2023-08-12
> > > **Response to further questions**
> > >
> > > Thank you for the further comments. Here are our responses:
> > >
> > > **1. Trajectories for the other experiments.**
> > >
> > > Thanks for your suggestion! We will add additional examples in the final version.
> > >
> > > **2. Why and how your method performs better than each of the baselines?**
> > >
> > > Compared to FNO which uses pre-simulated fixed data for training, MCNP can sample new initial fields per epoch, increasing the diversity of training data. As a result, MCNP can achieve similar or better results, especially when the PDE fields are varying at the final time for different initial fields, where more data are required for supervised methods. This can be seen in cases like the diffusion equation with $N=12$ (As discussed in Lines 253-255).
> > >
> > > Compared to PINO, MCNP is more robust against spatiotemporal variations due to the benefits of MCM (proved in Theorem 4.1 for convection-diffusion equation). Moreover, we propose the multi-scale framework to enhance the long-time simulation ability. Therefore, MCNP can outperform PINO significantly if PSM cannot simulate the PDE fields accurately with a relatively coarse time step, as in the cases with large spatiotemporal variations (such as diffusion equation with $\kappa=0.2$ and the Kolmogorov flow, discussed in Lines 255-256). Please note that we cannot make the time step in PINO sufficiently small due to the training cost.
> > >
> > > **3. FNO in the case $\nu=10^{-3}$.**
> > >
> > > For NSE with $\nu=10^{-3}$, the external forces make different initializations converge to the same final vorticity fields, which makes the final vorticity fields easier to learn. Moreover, the spatiotemporal variations are small in the low Reynolds number case. Therefore, the relative error for FNO is decreasing over time. On the other hand, unsupervised methods such as MCNP and PINO have an increasing error over time due to the numerical errors accumulated in each iteration.
> > >
> > > **4. The error metric in Table 2.**
> > >
> > > In Sec. 5.2, we conduct two kinds of experiments with different time ranges: $T\in[0,10]$ and $T\in[0,15]$. For each time range, we train neural PDE solvers separately and report the average relative error over time in Table 2. Therefore, the value of 6.553% in Table 2 corresponds to the average error from $t=0$ to $t=15$, while the value of ~14% in the figure is the relative error at $t=15$.
> > >
> > > Thanks for your question! We will clarify it in the final version.

---

> > > ### Author Response · Authors · 2023-08-20
> > > **Any Further Questions are Welcomed!**
> > >
> > > Dear Reviewer urJf,
> > >
> > > Considering the deadline of the current stage is approaching, we hope to know if there are any other concerns we haven’t addressed or any flaws in our rebuttal. We would be happy to discuss them with you and address them in future versions of the paper, which will also help us improve the quality of our work.
> > >
> > > Many thanks for your time and constructive comments!
> > >
> > > Sincerely,
> > >
> > > Paper 2692 Authors

---

> > > > ### Comment · Reviewer_urJf · 2023-08-20
> > > >
> > > > Thank you for your response. I've updated my score to reflect the improved presentation of the experiments. The experimental analysis is still the weakest part of the paper but the theoretical contribution is considerable.

---

### Official Review · Reviewer_R6Cs · 2023-07-06

**Soundness:** 3 good
**Presentation:** 2 fair
**Contribution:** 3 good
**Rating:** 6
**Confidence:** 2

**Summary:**

The authors propose Monte Carlo Neural PDE Solver (MCNP Solver) which leverages the Feynman-Kac formula to train neural PDE solvers in an unsupervised manner.

I'm willing to revise my score based on the rebuttal from the authors to the questions that I raised below.

**Strengths:**

* That paper addresses and interesting problem: learning the neural operator in an unsupervised way.
* The authors propose practical enhancements to their method such as one-step rollout, Fourier Interpolation and the use of a multi-scale framework.

**Weaknesses:**

* Limited set of experiments, only two cases: 1d diffusion and 2d Navier-Stokes.

**Questions:**

* Why aren't some other equations such as the Poisson equation, Schrodinger equation, Allen-Cahn not ran?
* To my understanding, you use the multi-scale framework to achieve longer simulations. Do you have an experiment that shows the benefit of this? Especially an experiment that shows how alternative method break at certain points whereas MCNP lasts longer.

**Limitations:**

* The limitations that I see are encapsulated on the questions that I raised above.

---

> ### Author Rebuttal · Authors · 2023-08-09
>
> Thank you for your time and valuable feedback! Based on your constructive comments, we provide some replies to address the weaknesses and questions:
>
> **Weakness & Q1. The PDE types in the experiments of this paper**
>
> Thank you for your question! Besides the 1D diffusion equation and 2D NSE, we also conduct additional experiments in the supplementary material (Sec. 5.4, Appendix C), including heat diffusion on a circular ring, fractional diffusion equations, and fractional diffusion equations with irregular grids. We choose the experiments according to the criterion that each experiment should convey new information to the readers. To clearly summarize our experiments, we list each experiment and corresponding insight as follows:
> 1. Diffusion equation (Sec. 5.1): echo our main motivation and theoretical result.
> 2. NSE (Sec. 5.2): a standard benchmark for neural PDE. We evaluate the ability of MCNP for handling large spatiotemporal variations and long-time simulation.
> 3. Heat diffusion on a circular ring (Appendix C.1): demonstrate the ability of MCNP to handle different boundary conditions.
> 4. Fractional PDEs (Appendix C.2): demonstrate the ability of MCNP to handle fractional Laplacian.
> 5. PDEs with irregular grids (Appendix C.3): demonstrate the ability of MCNP to handle irregular grids.
> 6. Kolmogorov flow (please refer to **Global Response B**): demonstrate the ability of MCNP to handle chaotic systems.
>
> The Poisson equation and Schrodinger equation do not fall within the scope of MCNP (Eq. 1), and some mathematical transformations are required. Therefore, we do not include these PDEs in the current version. Compared to the NSE, the Allen-Cahn equation is generally simpler and less representative in low-dimensional cases, and it may not provide new information about the characteristics of MCNP. Therefore, we choose NSE to demonstrate the performance of MCNP Solver, which is also a standard benchmark in most neural operator learning papers [1, 2].
>
> **Q2. The multi-scale framework**
>
> Thank you for your question! The ablation study of the multi-scale framework can be seen in Sec. 5.4, where MCNP-~~MS~~ denotes the MCNP Solver without the MS trick. We also add figures in the PDF file of the **Global Response B** part to show the variation of trajectories and errors for each neural PDE solver over time.
>
> We hope that our rebuttal can address your concerns. Also, we would like to know whether there are any other questions about our work, and we are happy to answer and discuss them. If the major weaknesses and questions have been addressed, could you please kindly raise the rating?
>
> [1] Li Z, Kovachki N, Azizzadenesheli K, et al. Fourier neural operator for parametric partial differential equations. ICLR, 2020.
>
> [2] Wu T, Maruyama T, Leskovec J. Learning to accelerate partial differential equations via latent global evolution. NeurIPS, 2022.

---

> > ### Author Response · Authors · 2023-08-18
> > **A further explanation to the effects of multi-scale framework**
> >
> > Dear Reviewer R6Cs,
> >
> > We would like to add a further explanation to clarify the effects of the multi-scale (MS) framework in our paper.
> >
> > In Section 5.4, we conducted the ablation study of the multi-scale framework to evaluate its effect, where MCNP-~~MS~~ denotes the MCNP without using the MS trick. To better address your concerns, we show the relative error of each step with MCNP and MCNP-~~MS~~ as follows:
> >
> > |Time|2|4|6|8|10|12|14|
> > | - | - | - | - | - |-|- |-|
> > |Error, MCNP|5.940$\pm$ 0.467|5.983$\pm$ 0.330|5.910$\pm$ 0.232|6.873$\pm$ 0.274|7.333$\pm$ 0.147|10.363$\pm$ 0.266|15.367$\pm$ 0.428|
> > |Error, MCNP-~~MS~~ |24.941$\pm$ 2.323|24.876$\pm$ 2.280|21.985$\pm$ 1.946|20.809$\pm$ 1.632|22.325$\pm$ 1.392|26.269$\pm$ 1.188|32.465$\pm$ 0.851|
> >
> > According to the above table, MCNP significantly outperforms MCNP-~~MS~~ throughout the entire time range, rather than after a certain time point. Therefore, the multi-scale framework can help the neural PDE solver obtain a more robust simulation during the whole time range for long-time simulation tasks. The potential reason for this has been discussed in our paper, _due to the independent parameterization and stop-gradient operator, the proposed multi-scale framework can prevent the prediction at time $t^{\prime}$ from producing harmful effects on the former time $t \leq t^{\prime}$ in the optimization stage_ (Lines 182-184).
> >
> > We hope our rebuttal and this new message can fully address your concerns. Also, if you have any other questions about our work, please do not hesitate to contact us.
> >
> > Sincerely,
> >
> > Paper 2692 Authors

---

> > > ### Comment · Reviewer_R6Cs · 2023-08-19
> > >
> > > I thank the authors for addressing my questions, especially in regards to the experiments.
> > >
> > > I have adjusted my score slightly.

---

### Official Review · Reviewer_xU1D · 2023-07-06

**Soundness:** 3 good
**Presentation:** 3 good
**Contribution:** 3 good
**Rating:** 6
**Confidence:** 3

**Summary:**

Designing neural PDE sovler using deep neural networks is a challenging task for which several solutions have been proposed in the literature using for instance networks that encode the initial conditions or physics informed neural networks.

The authors propose to use Monte Carlo methods to train neural PDE solver for the solution of a general convection-diffusion equation.
Using Feynman-Kacformula, the authors derive a loss function that can be used to learn a mapping that can simulate the target fields using the input parameters and the initial condition.

They propose a theoretical guarantee on the solution provided by the Monte Carlo solver and the paper illustrates the performance of the proposed method with a  one dimensional differential equation and a 2-dimensional Navier-Stokes equation.

**Strengths:**

The paper proposes a Monte Carlo based PDE solver strained via Monte Carlo approximation which can  handle  coarse time steps better than existing alternatives.

The proposed method does not require many particles in the numerical experiment and is computationally efficient in the settings explored.

Under some assumptions, the authors propose an upper bound on the error explicit in some hyperparameters of the approach in the case of a  convection diffusion equation.


**Weaknesses:**

Theorem 1 is obtained under several assumptions.
These assumptions should be discussed more, are they restrictive or common assumptions in the PDE literature ?

The claim that the approach is efficient even with few samples seems correct in the proposed experiments. However, these experiments are in dimension 1 and 2 and Monte Carlo methods can be cumbersome in high dimensional settings without tunning.
The authors could detail the explicit advice or theoretical guarantees we have for the error with respect to M.

The authors only explore one discretization scheme (Euler), the scheme used could have an impact on the performance of the method, can this be discussed ?

**Questions:**

The algorithm relies on many hyperparameters which appear in the upper bound of Th.1. The dependency on these hyperparameters could be discussed more. For instance, the authors claim that the third term in (15) can be controlled by the number of samples M, and that an excessive number of particles is not required in practice.
Is it possible to provide an explicit way to balance each term in the upper bound to guarantee a given precision ? How to choose M with respect to N or $\Delta t$ for instance ?

Section 4 focuses on a specific case where parameters are constant. Can the authors elaborate on the difficulties (practical and theoretical) when these parameters are not constant ?

In Section 3.3, Eq. (11), the authors propose to use an Euler Scheme to sample the stochastic process. Can the results be improved by choosing another discretization scheme ? Is this a sensitive step of the implementation ?
In comparison with PSM which requires to decrease $\Delta t$ to improve the upper bound (which is costly) the proposed method seems less sensitive and increasing M is enough to reduce the impact of the additional term. In terms of computational complexity,  increasing   M is not too intensive ?

The simulation study provides application in dimensions 1 and 2, have you any insight on how the method scales with d ?

**Limitations:**

The authors provide several research perspectives for this work.

---

> ### Author Rebuttal · Authors · 2023-08-09
>
> Thanks for your time and valuable feedback! According to your constructive comments, we make some replies to the weaknesses and questions:
>
> **W1. The assumption in Theorem.**
>
> The assumptions are reasonable for most cases and common in the PDE literature:
>
> - The solution can be expressed via Fourier basis.
>
> _Fourier series were historically developed in the analysis of classical PDEs in mathematical physics; these series were used to express the solution of such equations_ [1]. Moreover, many numerical methods assume the solution of PDE can be expressed by the Fourier basis, e.g., the algorithm and theoretical analysis in [2] and Theorem 2.1 in [3].
>
> - The solution and its derivatives are Lipschitz functions.
>
> Lipschitz assumptions are common in PDE literature. For example, paper [4] uses the property that the solution and derivatives are Lipschitz bounded in Lemma 9. Furthermore, most theorems in [5] rely on the Lipschitz assumption.
>
> We appreciate your feedback and will discuss these in the final version.
>
> **W2 & Q4. Experiments are in 1D and 2D.**
>
> We follow the common experimental setting in the research field of *Neural Operator*, and most of them consider the 1D and 2D PDEs. To the best of our knowledge, there are only very few papers that generalize the supervised operator learning methods to the 3D scenario [6], and no paper considers the unsupervised neural operator in the 3D scenario. To generalize MCNP to high-dimensional problems, we need to use some more powerful tools (like transformer) and regard it as an important future work.
>
> **W2, Q1 & Q3. Effect of M.**
>
> The theorem for MCM has another equivalent expression:
>
> - With probability at least $1-\gamma$, we have:
> $$
> |u_{t+\Delta t}^{MCM}(x) - u_{t+\Delta t}(x)| \leq \frac{1}{2H}\sum_{n=1}^N |na_n| + \sum_{n=1}^N |\delta_n| + \frac{\sqrt{4\kappa\Delta t}L_u^x}{\sqrt{M\gamma}}.
> $$
> Thank you for your suggestion. We will adopt this form in our final version. The theorem shows that the error term $\frac{\sqrt{4\kappa\Delta t}L_u^x}{\sqrt{M\gamma}}$ ($E_3$) can be controlled by the sampling number M. For MCM, we have to increase M to lower the error as $\Delta t$ and $\kappa$ increase. Please note this theorem is proved for MCM, and things are different for MCNP due to the use of neural networks. As discussed in the paper, $E_3$ _stems from the variance of random processes and can be regarded as a type of stochastic label noise. Some studies have found that such noise can aid generalization and counteract inherent biases._ To demonstrate this empirically, we add an experiment to reveal the effects of M on MCNP (the settings align with Sec. 5.1, $\kappa=0.02$).
>
> |M|32|64|128|256|
> |-|-|-|-|-|
> |Error, N=6|3.467$\pm$ 0.470|3.727$\pm$ 1.587|3.543$\pm$ 1.633|3.648$\pm$ 1.222|
> |Error, N=12|6.322$\pm$ 0.991|6.575$\pm$ 1.948|6.564$\pm$ 1.902|8.731$\pm$ 2.738|
>
> According to the table, we can see:
> 1. The results of MCNP are relatively robust to M.
> 2. M is not necessarily better when larger, which indicates that controlling the noise within a certain level can help generalization.
>
> Theoretically, to analyze the effects of M to MCNP more precisely, we need to consider the gradient flow during the training stage, and the main challenge is the non-convex property of neural networks. Therefore, we have acknowledged this limitation in the paper and regard it as future work.
>
> **W3 & Q3. Discretization scheme.**
>
> In this paper, we utilize the Euler–Maruyama method to approximate the SDE in Eq.11. Before that, we have tested other discretization schemes of SDE, including Runge–Kutta and Heun’s methods. However, these methods don’t give a significant improvement for MCNP while introducing remarkable computational costs. The results of these different discretization schemes are listed as follows (NSE data with $\nu=10^{-5}, T=15$):
>
> ||EM|RK|Heun|
> |-|-|-|-|
> |Error|8.667$\pm$ 0.350|8.648$\pm$ 0.266|8.621$\pm$ 0.318|
> |Time|1.458|2.162|1.971|
>
> To our best knowledge, other Feynman–Kac-based PINNs also don’t use high-order discretization schemes in their paper. One potential reason is these schemes may introduce extra optimization difficulties to the neural network and thus can not work as expected.
>
> **Q2. When parameters are not constant.**
>
> We have updated our theorem results in the **Global Response. A**. The main difficulty lies in the theoretical analysis. When $\beta$ is dependent on $x$, we need to estimate the error bound between the stochastic integral of ground-truth $\beta(x)$ and the simulated one $\beta(\hat{x})$ from $t$ to $t+\Delta t$. Please note that $\beta$ is involved in the random walk of $x$, and thus a composite structure arises. A rigorous proof requires more refined analysis and advanced mathematical tools in stochastic analysis.
>
> We hope our rebuttal can address your concerns. Also, we would like to know whether there are any other questions, and we are happy to answer and discuss them. If your major concerns have been addressed, could you please kindly raise the rating?
>
> [1] Plonka G, Potts D, et al. Numerical Fourier analysis. Basel: Birkhäuser, 2018.
>
> [2] Burns K J, Vasil G M, et al. Dedalus: A flexible framework for numerical simulations with spectral methods. Physical Review Research, 2020.
>
> [3] Gu Y, Shen J. An efficient spectral method for elliptic PDEs in complex domains with circular embedding. SIAM Journal on Scientific Computing, 2021.
>
> [4] Chassagneux J F, Crisan D, et al. Numerical method for FBSDEs of McKean–Vlasov type. The Annals of Applied Probability, 2019.
>
> [5] Kovachki N, Li Z, et al. Neural operator: Learning maps between function spaces with applications to PDEs. JMLR, 2023.
>
> [6] Peng W, Yuan Z, et al. Linear attention coupled Fourier neural operator for simulation of three-dimensional turbulence. PoF, 2023.

---

> > ### Comment · Reviewer_xU1D · 2023-08-16
> >
> > Thank you for your clarifications and for the answers to all reviewers, in particular for the additional experiments.
> >
> > I will update my ratings after discussions with other reviewers.

---

### Official Review · Reviewer_dyHf · 2023-07-26

**Soundness:** 3 good
**Presentation:** 3 good
**Contribution:** 2 fair
**Rating:** 5
**Confidence:** 4

**Summary:**

The paper proposes a new physics informed neural network based solver that utilizes the connection between PDEs and SPDEs. This is achieved through the Feynman-Kac formula and applies to a large class of PDEs. It comes with a bound on the error at each step in the rollout. The results are compared with multiple supervised and unsupervised PDE solvers on a number of 1D and 2D equations.

**Strengths:**

Originality: To the best of my knowledge, the paper is original in combining the Feynman-Kac-based approach with a neural operator architecture. The connections to existing work that relies on the Feynman-Kac formula as well as other PINN/NO approaches are discussed in detail in the main paper and the appendix.

Quality: The work is thorough in discussing existing literature and presenting the methodology. The theoretical result gives some intuition relating to how the proposed method scales as compared to the classical solver.

Significance: The methodology combines a number of existing techniques (Feynman-Kac formulation, neural operators, Fourier interpolation) to achieve some improvement in specific setups, e.g. where the solution of the PDE is rapidly varying in space and/or time.





**Weaknesses:**

Clarity: The quality of the writing could be improved, particularly in the abstract/introduction. The rest of the paper is detailed enough in describing the experiments and discussing the results.

Significance: The proposed approach seems to give an advantage only in specific situations. While this is acknowledged in the paper, it could be beneficial to discuss specific applications where such oscillatory conditions occur.



**Questions:**

1. What are FDM and PSM as mentioned in the abstract?

2. Is PSM a spectral solver?

3. Line 119: Can you explain in more detail what is meant by inversion of $\xi$?

4. What is the number of dimensions in the latent space (i.e. the number of terms in the Fourier expansion) for the FNO method in your experiments? I assume increasing the number of terms in the expansion would improve the performance of FNO for high-frequency initial states and might not affect the computational cost hugely.

5. To clarify, I assume the computational times given in Tables 1 and 2 do not include the generation of data for data-driven methods such as FNO?



**Limitations:**

The limitations and extent to which this method gives an advantage over existing approaches are discussed in detail in the final section of the paper.

---

> ### Author Rebuttal · Authors · 2023-08-09
>
> Thank you for your time and valuable feedback! According to your constructive comments, we make some replies to the weaknesses and questions:
>
> **W1: The quality of the writing could be improved.**
>
> Thank you for the valuable comments. We will take your suggestions on polishing the final version.
>
> **W2: The proposed approach seems to give an advantage only in specific situations.**
>
> Thank you for your comments. We need to clarify that each solver has its strengths and weaknesses, and we do not intend to claim that MCNP is superior to all baseline methods in all scenarios. Instead, we aim to comprehensively compare and show each method's advantages, disadvantages, and suitable scenarios. Furthermore, MCNP is trained in an unsupervised manner, while still obtaining comparable or even better results compared to the supervised FNO.
>
> In this paper, we tackle the challenging and crucial problem of solving PDEs with large spatiotemporal variations, which occur in many computational physics applications. For example, turbulence happens when an airplane meets unstable air currents in the sky [1]. Turbulence flow involves rapid multi-scale changes, which is a well-known Millennium Prize Problem and has significant implications for fluid physics and weather forecasting [2]. Therefore, developing robust numerical methods that can handle high spatiotemporal variations is a hot topic in both machine learning and computational physics communities [3,4,5].
>
> We appreciate your feedback and will highlight the significance of our problem setting in the final version.
>
>
> **Q1 & Q2. The meaning of FDM and PSM.**
>
> FDM and PSM stand for the finite difference and pseudo-spectral methods, respectively. We will explain them in more detail in the final version. Thank you for your suggestions!
>
> **Q3. Inversion of $\xi$ (Line 119).**
>
> Thanks for your question. In formular, $\xi_{s} = \tilde{\xi}_{T-s}$ represents the  time inversion random process of $\tilde{\xi}$.
>
> **Q4. The number of terms in the Fourier expansion for FNO.**
>
> Thanks for your question! We provide the details of the experimental settings in Appendix E. We choose the number of terms in the Fourier expansion (i.e., the modes) from the set {12,16,20,24} on the most challenging task when conducting FNO, i.e., the highest frequency task. For the diffusion equation and NSE, we select modes 20 and 16, respectively. Then, we align the modes in PINO and MCNP with those in FNO.
>
> **Q5. The computational times given in the Tables do not include the generation of data for FNO?**
>
> Yes, you are right. We will clarify it in the final version.
>
> We hope that our rebuttal can address your concerns. Also, we would like to know whether there are any other questions about our work, and we are happy to answer and discuss them. If the major weaknesses and questions have been addressed, could you please kindly raise the rating?
>
> [1] Gerogiannis V T, Feidas H. An 11-year analysis of in situ records of aviation-scale turbulence over Europe. Theoretical and Applied Climatology, 2021.
>
> [2] Wilcox D C. Turbulence modeling for CFD. La Canada, CA: DCW industries, 1998.
>
> [3] Krishnapriyan A, Gholami A, Zhe S, et al. Characterizing possible failure modes in physics-informed neural networks. NeurIPS, 2021.
>
> [4] Li X A, Xu Z Q J, Zhang L. A multi-scale DNN algorithm for nonlinear elliptic equations with multiple scales. Communications in Computational Physics, 2020.
>
> [5] Schaeffer H, Caflisch R, Hauck C D, et al. Sparse dynamics for partial differential equations. Proceedings of the National Academy of Sciences, 2013.

---

> > ### Comment · Reviewer_dyHf · 2023-08-18
> >
> > Thank you for the responses.

---

### Author Rebuttal · Authors · 2023-08-09

## **Global Response**

**A. Errata of the main Theorem**

We found a typo in the main theorem, and we fix it as follows:

- The convection-diffusion equation in Eq. 13 should be:
$$
\frac{\partial u}{\partial t} = \kappa \Delta u + \beta \frac{\partial u}{\partial x},
$$
where $\beta \frac{\partial u}{\partial x}$ and $\kappa \Delta u$ denote the convection and the diffusion term, respectively.

- The bound and proof of MCM don't need to be corrected, and the error bound of PSM should be corrected as follows due to the involvement of the convection term:

$$
\left|u_{t+\Delta t}^{\operatorname{PSM}}(x) - u_{t+\Delta t}(x)\right| \leq \sum_{n=1}^N \frac{(|\kappa L_{\Delta u}^{t}| + |\beta L^t_{\partial_x u}|) {\Delta t}^2}{2}+\sum_{n=1}^N (|\delta_n(\kappa n^2 \Delta t - 1)| + |\beta n \Delta t\delta_n|),
$$

The new error term is introduced due to the error of the convection term.

**B. New experimental results**

In response to the reviewers’ request, we add some new numerical results as follows:

**B.1. NSE with Kolmogorov forcing.**

Apart from the force term introduced in Eq. 17, we also add an experiment to simulate NSE with Kolmogorov forcing [1]. We set the external forcing as $f(x) = 0.1\cos(8\pi x_1)$ and the viscosity term as $10^{-4}$. Other settings are in line with the ones in Sec. 5.2. The performances of all methods are presented in the following table:

|Method|PSM|PSM+|FNO|PINO|MCNP|
| - | - | - | - | - |-|
|Error, T=10|NAN|0.222|5.050$\pm$ 0.081|8.806$\pm$ 0.240|7.232$\pm$ 0.100|
|Error, T=15|NAN|0.319|9.738$\pm$ 0.219|26.250$\pm$ 0.608|10.747$\pm$ 0.346|

Please note that PSM and PSM+ are traditional numerical solvers, FNO is the supervised neural operator, and PINO and MCNP are trained in an unsupervised manner. We will integrate this new result into Table 2 of the original paper.

**B.2. Simulated trajectories for each neural PDE solver.**

In Fig. 1 of the attached PDF file, we show the ground-truth vorticity versus the prediction of learned neural solvers for an example in the test set from $t=3$ to $t=15$, with the viscosity terms $\nu=10^{−4}$. For both two forcings, the unsupervised MCNP obtains comparable simulation results compared to the supervised methods FNO. Furthermore, PINO fails to capture the details and trends of the fluid fields when $T\geq9$.

**B.3. Variation of errors for each neural PDE solver.**

In Fig. 2 of the attached PDF file, we compare the relative error of each time step with different neural PDE solvers. We summarize our observations as follows:
1. PINO fails to simulate NSE with $\nu≤10^{−4}$, where the fluid field changes drastically, and thus learning the subsequent vorticity field could have a bad effect on the front one for PINO.
2. MCNP obtains comparable results on most tasks compared to the supervised method FNO when $t\leq10$, and FNO can give a more precise prediction for $t>10$. As discussed in our paper, _FNO directly uses the ground-truth data as training labels for all $t\in[0,T]$, thus avoiding accumulated errors arising from the calls of the solver during the training stage like other unsupervised methods._ Moreover, when FNO simulating NSE with $\nu=10^{−3}$, we notice that the relative error even decreases over time, a possible reason is that in the low Reynolds number case, due to the presence of external forces, different initializations tend to have the same final vorticity fields, which is more convenient for the learning of supervised method.
3. For Kolmogorov flow, MCNP and FNO have similar performance during $t\in[0,15]$. The reason is that the final fields are very different among the datasets due to the involvement of Kolmogorov forcing, thus the supervised methods need more data to achieve a good performance.

**C. Code**

In response to the reviewers’ request, we decided to publish our code immediately. According to the NeurIPS review policy, _If you were asked by the reviewers to provide code, please send an anonymized link to the AC in a separate comment (make sure the code itself and all related files and file names are also completely anonymized)_.

We have submitted the code link to the AC and the code will be public once the paper is accepted.

[1] Smaoui N, El-Kadri A, Zribi M. On the Control of the 2D Navier–Stokes Equations with Kolmogorov Forcing. Complexity, 2021.

---

### Author Response · Authors · 2023-08-21
**A Summary of Rebuttal Stage and the Revision Plan**

Dear Reviewers and AC,

We sincerely thank all the reviewers and AC for handling our paper and for this great discussion. According to the reviews and rebuttals, we summarize the rebuttal stage and our revision plan as follows:

**Theory**

- **(Have Done)** Fix the typo in the main Theorem (please refer to **Global Response A**).

- **(Have Done)** Use another equivalent expression to state the theorem, as discussed with Reviewer `xU1D`.
- **(Have Done)** Clarifying how to label noise could help generalization in more detail, as discussed with Reviewers `xU1D` and `urJf`.

**Experiments**

- **(Have Done)** Publish our code, which has already been submitted to AC.

- **(Have Done)** Integrate the additional experiments conducted during the rebuttal period in the final version, including Kolmogorov flows, the effects of $M$, and the discretization scheme, as suggested by Reviewers `xU1D`, `R6Cs`, and `urJf`.

- **(Partly Done)** Add more detailed and comprehensive experimental analyses and show more trajectories for various methods, as discussed with Reviewers `urJf`, `R6Cs`, and `MGTF`.

**Writing**

- **(Have Done)** Clarify some abbreviations and misunderstandings, as suggested by Reviewers `dyHf`, `urJf`, `R6Cs`, `xU1D` and `MGTF`.

- **(Have Done)** Emphasise the significance of the target setting of our paper, as suggested by Reviewer `dyHf`.

- **(Partly Done)** Polish the abstract and introduction part in the final version, as suggested by Reviewer `dyHf`.

**Last but not least**, we are still waiting for Reviewer `MGTF`’s response, though we failed to receive any comments. We hope to know if there are any other concerns that we haven't addressed and if Reviewer `MGTF` is satisfied with our rebuttal and revision plan.

We will revise this paper as we promised, no matter whether it is accepted or rejected. Many thanks for all reviewers' and AC's time and constructive comments!

Sincerely,

Paper 2692 Authors

---

### Decision · Program_Chairs · 2023-09-21

**Decision:**

Reject

**Comment:**

This is an interesting paper that, however, is borderline with many reviewers weakly recommending accepting the paper and some arguing for rejection. From the discussion, the reviewers see value in this paper and found the introduced method interesting and new in the context of deep learning. The method itself along with the theoretical results are strong points. However, the experiments lacked both in breadth and depth. The reviewers also recommend adding additional details regarding the background of the proposed method and highlighting the settings where the proposed methodology is advantageous. This is to make the paper more assessable and ensure proper impact.